# mTORC1 upregulates B7-H3/CD276 to inhibit antitumor T cells and drive tumor immune evasion

Heng-Jia Liu ⬢[1] ✉, Heng Du[1], Damir Khabibullin[1], Mahsa Zarei[1,2], Kevin Wei[3], Gordon J. Freeman ⬢[4], David J. Kwiatkowski ⬢[1] & Elizabeth P. Henske ⬢[1] ✉

Identifying the mechanisms underlying the regulation of immune checkpoint molecules and the therapeutic impact of targeting them in cancer is critical. Here we show that high expression of the immune checkpoint B7-H3 (*CD276*) and high mTORC1 activity correlate with immunosuppressive phenotypes and worse clinical outcomes in 11,060 TCGA human tumors. We find that mTORC1 upregulates B7-H3 expression via direct phosphorylation of the transcription factor YY2 by p70 S6 kinase. Inhibition of B7-H3 suppresses mTORC1-hyperactive tumor growth via an immune-mediated mechanism involving increased T-cell activity and IFN-γ responses coupled with increased tumor cell expression of MHC-II. CITE-seq reveals strikingly increased cytotoxic CD38[+]CD39[+]CD4[+] T cells in B7-H3-deficient tumors. In pan-human cancers, a high cytotoxic CD38[+]CD39[+]CD4[+] T-cell gene signature correlates with better clinical prognosis. These results show that mTORC1-hyperactivity, present in many human tumors including tuberous sclerosis complex (TSC) and lymphangioleiomyomatosis (LAM), drives B7-H3 expression leading to suppression of cytotoxic CD4[+] T cells.

Tumor cells escape immune surveillance via multiple mechanisms[1]. One of these mechanisms is the induction of endogenous immune checkpoints that normally terminate immune responses after antigen activation[2]. For example, programmed cell death ligand 1 (PD-L1/B7-H1), a B7 family immune checkpoint molecule, can be upregulated transcriptionally, post-transcriptionally, and post-translationally in tumors[3,4]. B7 homology 3 (B7-H3), encoded by the *CD276* gene, shares 28% and 29% amino acid identity with PD-L1/B7-H1 and PD-L2/B7-DC, respectively[5,6]. B7-H3 plays both immunological and non-immunological roles and can be co-stimulatory or co-inhibitory depending on cell type and context. In models of cardiac allograft rejection, encephalomyelitis, and airway inflammation, B7-H3 promotes inflammatory responses and enhanced proinflammatory

cytokine and chemokine production[7–9]. B7-H3 can also act as a T cell co-stimulatory protein in vitro and in autoimmune disease models[6,8]. In contrast, B7-H3 suppresses antitumor immunity in cancer models by reducing the cytotoxic activity of CD8[+] T cells and natural killer (NK) cells[10–13] and can also promote migration, invasion, and metastasis, independent of the immune system[14–16]. Despite these advances, how B7-H3 remodels the cellular composition and transcriptional landscape of the tumor immune microenvironment has not been comprehensively examined.

High levels of B7-H3 expression are found in many human malignancies[17–22] and higher levels of B7-H3 are associated with poor prognosis[23–27]. Tumor-associated vasculature and fibroblasts can also have elevated B7-H3 expression[17,28]. Expression of B7-H3 in normal

[1]Pulmonary and Critical Care Medicine, Department of Medicine, Brigham and Women's Hospital and Harvard Medical School, Boston 02115 MA, USA. [2]Department of Veterinary Physiology and Pharmacology, College of Veterinary Medicine and Biomedical Sciences, Texas A&M University, College Station 77843 TX, USA. [3]Division of Rheumatology, Inflammation, and Immunity, Brigham and Women's Hospital and Harvard Medical School, Boston 02115 MA, USA. [4]Department of Medical Oncology, Dana-Farber Cancer Institute, Harvard Medical School, Boston 02215 MA, USA. ✉e-mail: hliu1@bwh.harvard.edu; ehenske@bwh.harvard.edu

human and mouse tissues is generally low[17,29]. Perhaps because of this low expression in normal tissues, therapies targeting B7-H3, which are currently in clinical trials, have shown no dose limiting toxicities. Although microRNAs (miR-29 and miR-124, miR-128), immunoglobulin-like transcript (ILT) 4, and more recently autophagy have been implicated as B7-H3 regulators[23,30-33], the underlying mechanisms through which B7-H3 is upregulated in tumors, and how B7-H3 impacts their escape from immunosurveillance, remain largely unknown.

The mechanistic/mammalian target of rapamycin complex I (mTORC1) is a central regulator that coordinates eukaryotic cell growth and metabolism with environmental stimuli, including growth factors and nutrients[34-37]. The Tuberous Sclerosis Complex proteins TSC1 (hamartin) and TSC2 (tuberin) are key negative regulators of mTORC1 and play a critical role in tumor suppression[38,39].

Here, we demonstrate that high mTORC1 activity together with *CD276* expression is associated with poor overall survival and immunosuppressive phenotypes in The Cancer Genome Atlas (TCGA) pan-cancer cohort (over 10,000 patients, across 34 different cancer subtypes). B7-H3 expression is dependent on the transcription factor YY2 which is phosphorylated by p70 S6 kinase, a direct substrate of mTORC1. B7-H3 inhibition in mTORC1-hyperactive cells suppresses tumor growth by increasing IFN-γ responses and major histocompatibility complex II (MHC-II) expression in the tumor cells and enhancing CD38+CD39+CD4+ T lymphocyte-mediated antitumor immunity. These results reveal important fundamental mechanisms of B7-H3 upregulation and its impact on human tumors, with special relevance to tumors with mutations in the *TSC1/2* genes, including tuberous sclerosis complex (TSC)-associated tumors of the brain, heart, and kidney and pulmonary lymphangioleiomyomatosis (LAM).

## Results

### High B7-H3 expression and high mTORC1 signatures associate with poorer prognosis and less antitumor immune cell infiltrates

To examine B7-H3 expression and mTORC1 activation in cancer, we analyzed the expression profiles of more than 10,000 tumors (from 34 different cancer types) from The Cancer Genome Atlas (TCGA) database. We found that High *CD276* expression associated with lower overall survival times (Fig. 1a; $P < 0.0001$). We then calculated an mTORC1 activity score for each patient by performing a single-sample GSEA (ssGSEA) with TCGA-derived datasets using 199 mTORC1 signaling genes from the Molecular Signatures Database (MSigDB). We found that tumors have a high mTORC1 score have significantly lower overall survival times (Fig. 1b; $P < 0.01$). We then grouped tumors into high or low mTORC1 score and high or low *CD276* expression. A high mTORC1 score and high *CD276* expression have a significant negative correlation with overall patient survival (Fig. 1c, $P < 0.0001$). To meet proportional hazards, we selected the 11 cancer types with death events greater than 20 from patients with high and low mTORC1 score and high and low *CD276* expression. The combined hazard ratio (HR) for these 11 cancer types evaluating *CD276* expression is 1.22 (95% CI: 1.06-1.42, $P < 0.001$) and together with the mTORC1 score is 1.23 (95% CI: 0.99-1.53, $P = 0.058$), suggesting that the combined *CD276* expression and mTORC1 score is an indicator of poor prognosis (Fig. 1d-f).

To determine whether B7-H3 is associated with mTORC1 activity, we analyzed the protein expression of B7-H3, phospho-S6 (pS6) and phospho-S6K (pS6K) in 130 tumors representing 12 different tumor types. B7-H3 expression positively correlates with the levels of pS6 (Fig. 1g, h; $p = 0.0018$) and pS6K (Fig. 1i, j; $p = 0.0475$). Together, high mTORC1 score, high *CD276* expression, or both with the aggregated patterns across cancer types.

We next asked whether mTORC1 and *CD276* are associated with immune cell signatures in cancer by analyzing the pan-TCGA data using the 6 core immune subtypes (C1-C6) identified by Thorsson et al.[40] mTORC1 scores appear to be similar in all 6 immune subtypes

(Fig. 1k). Low CD276 expression generally falls in the C5 immunologically quiet category and high *CD276* falls in C6-TGF-β dominant (defined by the highest TGF-β signature and high lymphocytic infiltrate) category (Fig. 1l). These tumors with high mTORC1 score and high *CD276* expression display an immunosuppressive phenotype, with fewer CD8+ T cells and activated NK cells plus higher numbers of Tregs and M2 macrophages (Fig. 1m).

### B7-H3 expression is dependent on mTORC1 but not mTORC2

Having found that mTORC1 activity is correlated with B7-H3 expression in human tumor specimens (Fig. 1g, h), we next sought to identify the upstream molecular mechanisms governing B7-H3 expression. As a model of mTORC1 hyperactivity, we first examined B7-H3 expression in three TSC2-deficient cell lines: 621-101 cells, derived from a benign renal angiomyolipoma with a TSC2 mutation (G1832A or R611Q) and TSC2 loss of heterozygosity (LOH) at chromosome 16, 105K cells, derived from a renal tumor that developed in a *Tsc2*+/− mouse, and Tsc2 KO mouse embryonic fibroblasts, generated from Rosa26Cre-ER^T2^/ *Tsc2*^flox/flox^ mice. In all three models, elevated levels of B7-H3 protein (Fig. 2a-c) and *CD276* mRNA (Fig. 2d-f) were seen relative to TSC2-expressing controls (Fig. 2a-c). Treatment with the mTOR inhibitors Rapamycin (which inhibits mTORC1) or Torin 1 (which inhibits mTORC1 and mTORC2) for 24 hours reduced B7-H3 protein expression (Fig. 2g-i) and mRNA expression (Fig. 2j-l) in the TSC2-deficient cells. Treatment of A549 (lung cancer), T47D (breast cancer) and PC3 (prostate cancer) cells with Rapamycin or Torin 1 similarly suppressed the protein and mRNA expression of B7-H3 (Supplementary Fig. 1a-f), further validating that mTORC1 is a critical regulator of B7-H3 expression.

To determine whether mTORC1, mTORC2, or both regulate B7-H3, we used siRNA to downregulate Raptor, Rictor, mTOR, S6K, or 4E-BP1 in *Tsc2* wild-type and *Tsc2* KO MEFs. Downregulation of the mTORC1 components (Raptor, mTORC1), or its downstream target S6K reduced B7-H3 protein (Fig. 2m) and mRNA levels (Fig. 2n) in *Tsc2* KO MEFs, but not *Tsc2* wild-type MEFs. Similar results were seen in Tsc2-deficient 105K kidney tumor cells (Supplementary Fig. 1g, h). siRNA downregulation of Rictor did not impact B7-H3 levels (Fig. 2m, n). We also utilized MEFs with inducible loss of Raptor (referred to as iRapKO) or Rictor (referred to as iRicKO) to confirm the siRNA results. Deletion of Raptor, but not Rictor, reduced B7-H3 levels (Supplementary Fig. 1i, j). These data demonstrate that mTORC1 regulates the expression of B7-H3.

We investigated how activating mTORC1 mutations impact B7-H3 expression by expressing 4 different activated mTORC1 mutants in 293 T cells. Expression of all 4 mutants increased B7-H3 expression compared to wild-type mTORC1 (Supplementary Fig. 1k). To further understand how B7-H3 is regulated in a more natural context of mTOR activation, we stimulated wild-type mouse embryonic fibroblasts with amino acids to activate mTORC1 and examined B7-H3 protein expressing. We found that amino acid stimulation leads to increased B7-H3 expression (Supplementary Fig. 1l).

### CD276 is transcriptionally controlled by Yin-Yang 2 (YY2)

We next focused on the transcriptional regulation of *CD276* mRNA. The full-length 5′ UTR region of *Cd276*, which contains its promoter sequences, was cloned into a luciferase reporter plasmid. *Cd276* promoter activity was higher in Tsc2-deficient 105K cells and *Tsc2* KO MEFs, compared with controls (Fig. 3a, b). The promoter activity was suppressed by inhibition of Raptor, mTOR, and S6K, as expected based on the results in Fig. 2 (Fig. 3c, d).

To identify the transcription factors that bind to the *CD276* promoter region, we input mouse and human *CD276* promoter sequences in Genomatix MatInspector software (https://www.genomatix.de) and later confirmed by JASPAR[41], leading to the identification of three candidates: STAT3, YY1 and YY2. Knockdown of STAT3 or YY1 did not

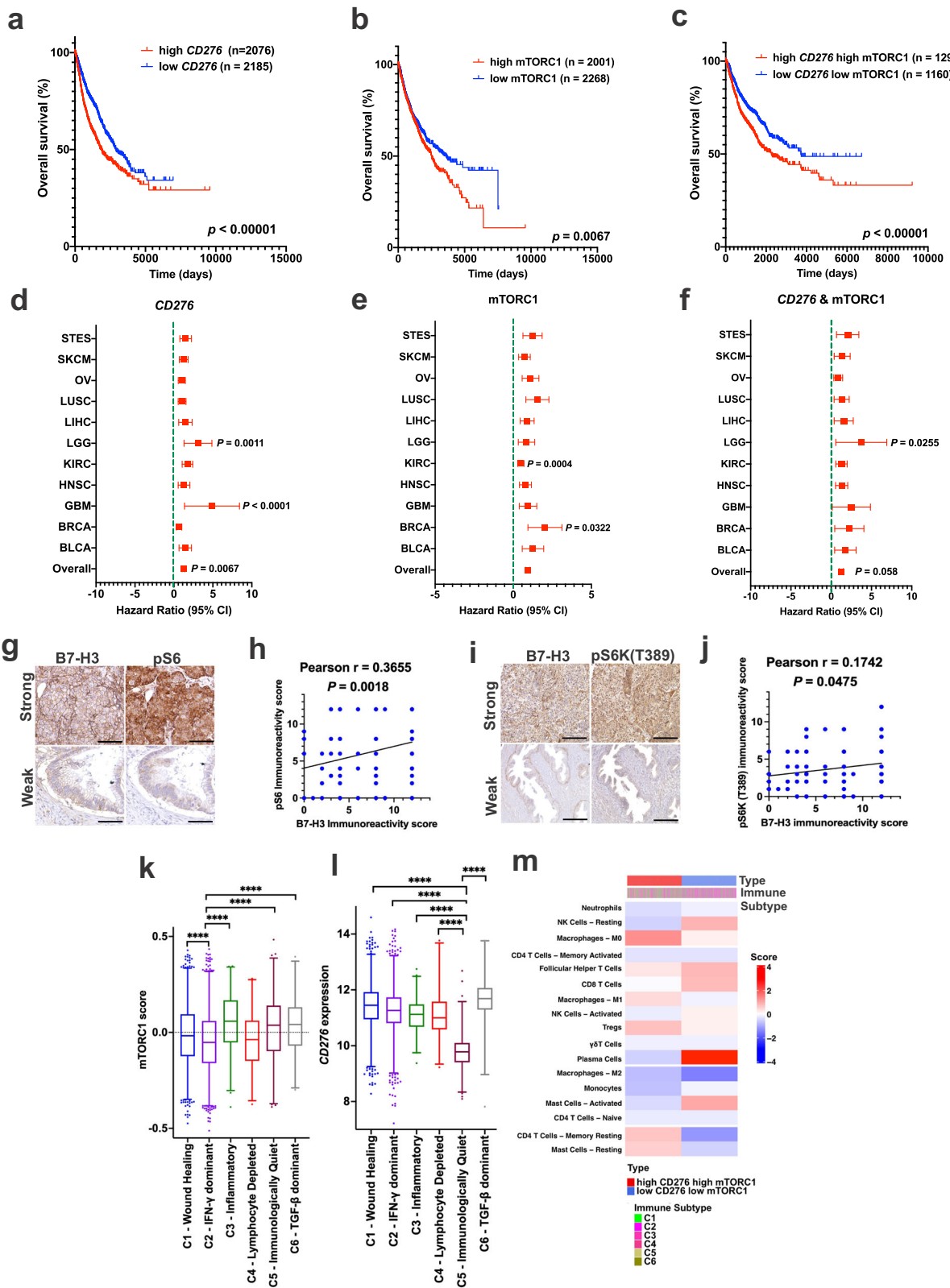

impact B7-H3 expression in Tsc2-deficient 105K cells or *Tsc2* KO MEFs (Supplementary Fig. 1m–o), but YY2 knockdown reduced B7-H3 protein and mRNA expression by ~50% (Fig. 3e–h). The *CD276* promoter region (both human and mouse) has a YY2 binding motif (atctgtC-CATtacacagacagga, where CCAT is the YY2 anchor motif) at nucleotides 479–501 before the transcriptional start site (Fig. 3i). Chromatin immunoprecipitation (ChIP)-qPCR confirmed the binding of YY2 to

*Cd276* promoter is higher in Tsc2-deficient cells, with a 6-fold ($p < 0.01$) increase in Tsc2-null 105K and a 2-fold ($p < 0.01$) in *Tsc2* KO MEFs compared to controls (Fig. 3j, k). Downregulation of YY2 using siRNA suppressed CD276 promoter activity by <50% in both Tsc2-deficient 105K cells and *Tsc2* KO MEFs (Fig. 3l, m). Together, these data indicate that YY2 upregulates CD276 transcription in cells with hyperactive mTORC1.

**Fig. 1 | Elevated mTORC1 activity and B7-H3/*CD276* expression are associated with poorer clinical outcomes and immunosuppressive phenotypes.** Kaplan-Meier plot of overall patient survival stratified by high ($n = 2076$) and low ($n = 2185$) *CD276* expression (**a**), high ($n = 2001$) and low ($n = 2268$) mTORC1 score (**b**), and high *CD276* expression plus high mTORC1 score ($n = 1296$) and low *CD276* expression plus low mTORC1 score ($n = 1160$) (**c**). Log-rank test. Forest plot of hazard ratios by tumor type and overall (with 95% confidence intervals) stratified by high and low *CD276* expression (**d**), high and low mTORC1 score (**e**), and high and low *CD276* expression plus high and low mTORC1 score (**f**). Hazard ratio greater than 1 (red) indicates an association with worse outcome. COX proportional hazards regression with Efron's approximation. **g**, **h** Representative images of strong and weak immunohistochemical staining of phospho-S6 (Ser240/244) (pS6) and B7-H3 in tumor tissue microarrays, which include 12 cancer types: meningothelial meningioma, breast infiltrating ductal adenocarcinoma, colon adenocarcinoma, stomach adenocarcinoma, prostate adenocarcinoma, uterus adenocarcinoma, ovarian adenocarcinoma, esophageal squamous cell carcinomas, kidney clear cell carcinomas, hepatocellular carcinomas, lung adenocarcinoma/squamous cell carcinomas, and bladder transitional/papillary cell carcinomas. Scale bar = 50 μm (**g**). Correlation analysis between the phospho-S6 immunoreactivity score and B7-H3 expression score in the tumors in **g** (Pearson correlation coefficient test) (**h**). **i**, **j** Representative images of strong and weak immunohistochemical staining of phospho-S6K (Thr389) (pS6K) and B7-H3 in tumor tissue microarrays, which include 12 cancer types indicated in (**g**). Scale bar = 50 μm (**i**). Correlation analysis between the phospho-S6K immunoreactivity score and B7-H3 expression score in the tumors in **i** (Pearson correlation coefficient test) (**j**). mTORC1 score (**k**) and *CD276* expression (**l**) among the six immune subtypes. Bars indicate median line, boxes denote interquartile range, whiskers denote 1–99 percentile and outliers are shown. Nonparametric Kruskal-Wallis test with Dunn's multiple comparisons test. $n = 5429$, ****$p < 0.0001$. **m** Heatmap indicates mean scores of immune-cell of all patients in high *CD276* and high mTORC1 tumors (left) and low *CD276* - low mTORC1 tumors (right). Source data is provided in the Source data file.

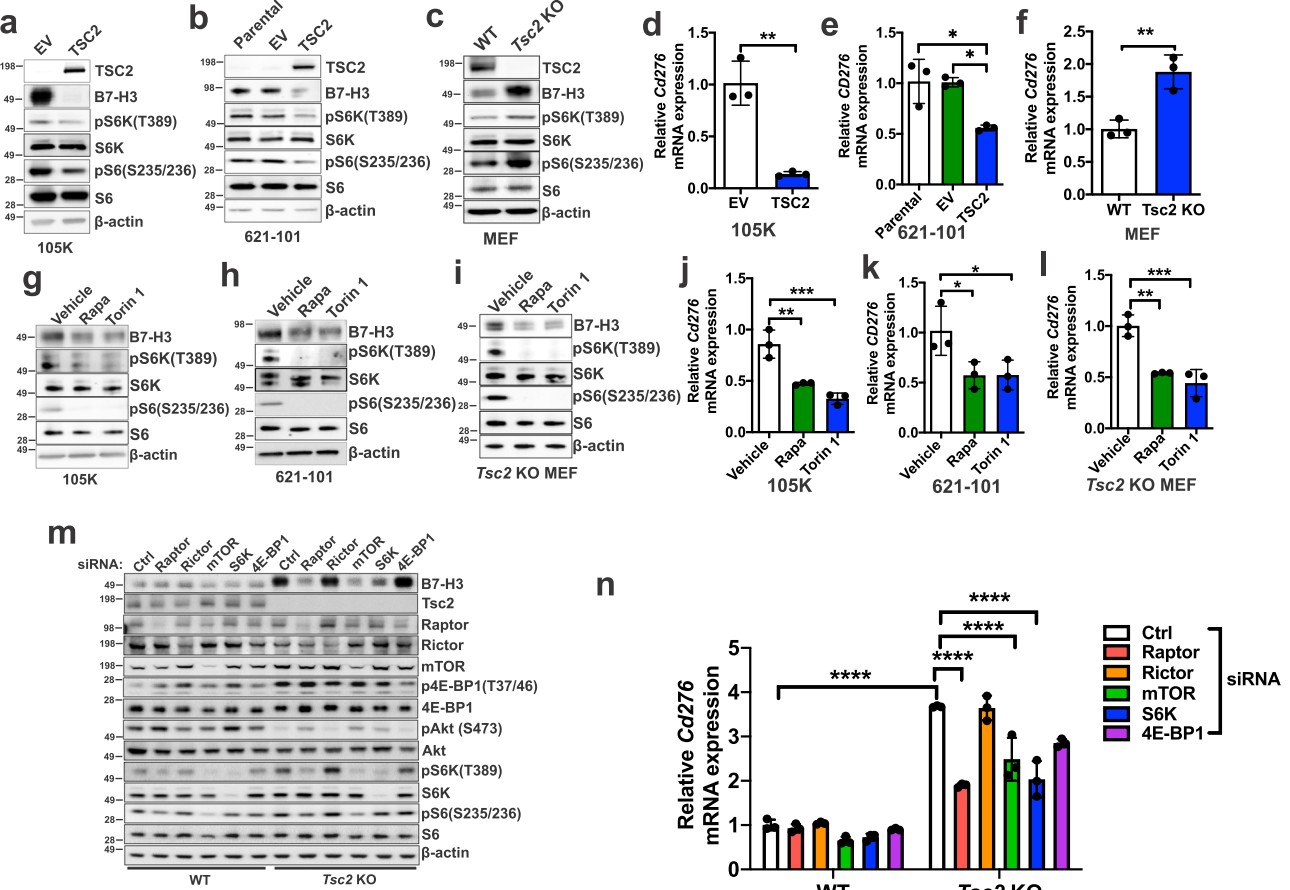

**Fig. 2 | B7-H3 expression is regulated by mTORC1.** B7-H3 protein expression in *Tsc2*−/− 105K cells with stable reconstitution of TSC2 or empty vector (EV) (**a**), *TSC2*−/− 621-101 angiomyolipoma-derived cells with reconstitution of TSC2 or empty vector (EV) (**b**), and *Tsc2*-WT and *Tsc2* KO MEFs (**c**). For all figure legends, $n = 3$ indicates representative of 3 biologic samples ($n = 3$). B7-H3 mRNA expression in *Tsc2*−/− 105K cells with stable reconstitution of TSC2 or empty vector (EV) (**d**), *TSC2*−/− 621-101 angiomyolipoma tumor cells with reconstitution of TSC2 or empty vector (EV) (**e**), *Tsc2*-WT and *Tsc2* KO MEFs (**f**). Means ± SD, two-tailed unpaired Student's *t*-test (**d**, **f**) or one-way ANOVA with Dunnett's multiple comparisons test (**e**), *$p < 0.05$, **$p < 0.01$. $n = 3$. B7-H3 protein expression in *Tsc2*−/− 105K cells (**g**), *TSC2*−/− 621-101 angiomyolipoma tumor cells (**h**), and *Tsc2* KO MEFs (**i**) treated with 20 nM rapamycin (Rapa), 500 nM Torin 1 or vehicle for 24 hr ($n = 3$). B7-H3 mRNA expression in *Tsc2*−/− 105K cells (**j**), *TSC2*−/− 621-101 angiomyolipoma tumor cells (**k**), and *Tsc2* KO MEFs (**l**) treated with 20 nM rapamycin (Rapa), 500 nM Torin 1, or vehicle for 24 hr. $n = 3$, means ± SD, one-way ANOVA with Dunnett's multiple comparisons test, *$p < 0.05$, **$p < 0.01$, ***$p < 0.001$. **m** Immunoblotting analysis of *Tsc2*-WT and *Tsc2* KO MEFs transfected with non-targeting control siRNA (Ctrl) or SMARTpool siRNAs targeting Raptor, Rictor, mTOR, S6K, or 4E-BP1 for 48 h hr ($n = 3$). **n** qRT-PCR analysis of *Tsc2*-WT and *Tsc2* KO MEFs transfected with non-targeting control siRNA (Ctrl) or SMARTpool siRNAs targeting either Raptor, Rictor, mTOR, S6K, or 4E-BP1 for 48 hr. $n = 3$, means ± SD, two-way ANOVA with Dunnett's multiple comparisons test, ***$p < 0.001$, ****$p < 0.0001$. Source data and exact $p$ values are provided in the Source data file.

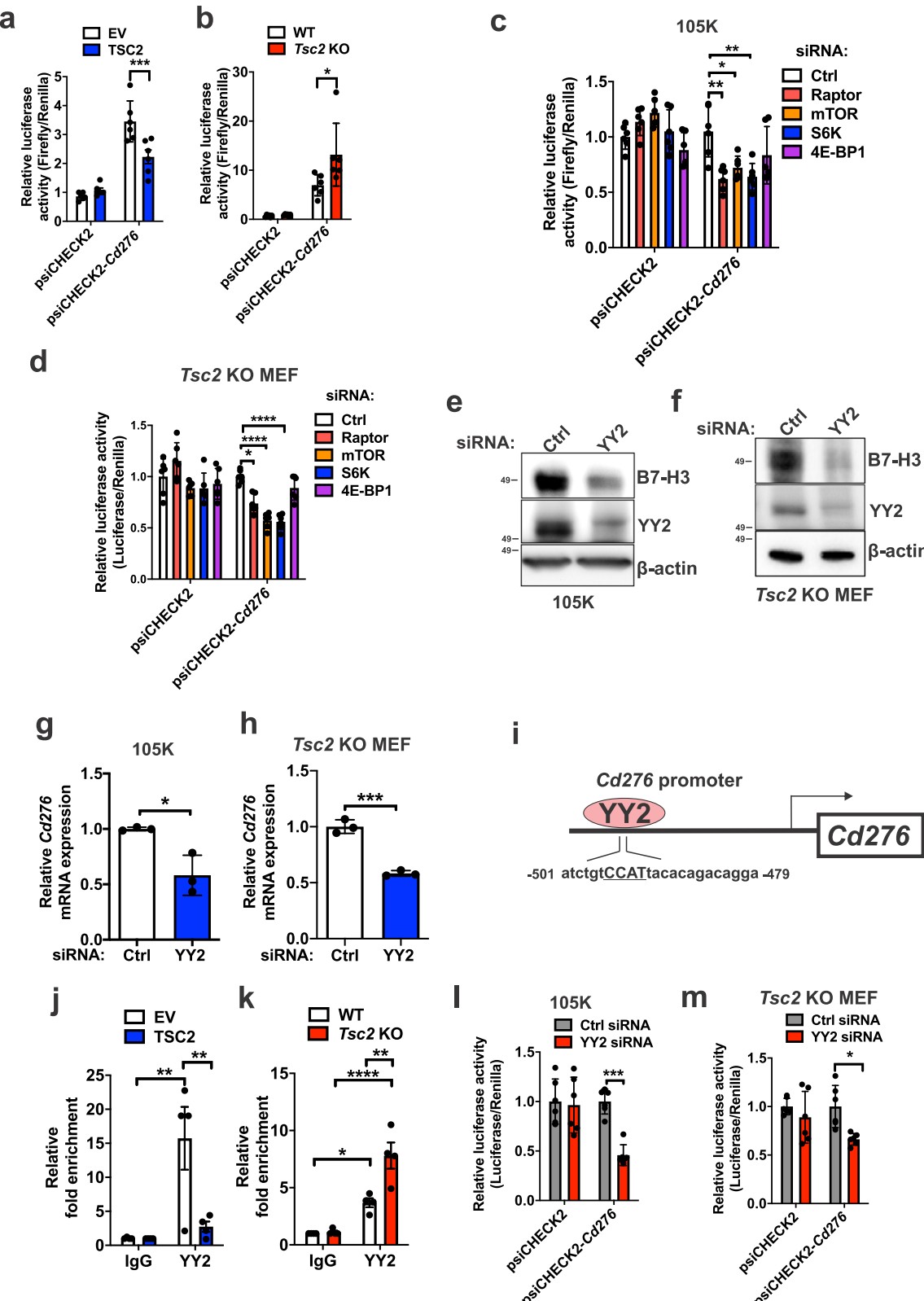

### YY2 is phosphorylated and stabilized by S6K via Smurf1

Having found that YY2 binds and promotes CD276 transcription and that CD276 promoter activity is dependent on S6K (Fig. 3c–m), we next sought to determine whether YY2 expression is mTORC1 dependent. We treated Tsc2-deficient 105K and *Tsc2* KO MEFs with Rapamycin and found YY2 protein expression was reduced by ~50% in both cell lines (Fig. 4a, b); the mRNA expression was unchanged (Supplementary

Fig. 2a, b). *Yy1*'s mRNA was also unchanged (Supplementary Fig. 2c, d). Treatment with an S6K1 inhibitor (PF-4708671) also reduced YY2 protein levels by ~50% (Fig. 4c, d). Cell fractionation revealed reduced YY2 levels in the nucleus of *Tsc2* KO MEFs following 2 hours of PF-4708671 (Supplementary Fig. 2e). This was confirmed by immuno-fluorescence showing Rapamycin, Torin 1 or PF-4708671 treatment does not affect YY2 translation to the cytosol (Supplementary Fig. 2f).

**Fig. 3 | YY2 controls CD276 transcription in Tsc2-deficient cells.** Enhanced *Cd276* promoter activity in *Tsc2−/−* 105K cells expressing empty vector (EV) compared to reconstitution of TSC2 (**a**) and in *Tsc2* KO MEFs compared to *Tsc2*-WT MEFs (**b**). Relative luciferase activity was determined by a dual-luciferase assay system. psiCHECK2-Cd276 encodes the *Cd276* promoter. Empty psiCHECK2 vector was used as the negative control. $n = 6$, means ± SD, two-way ANOVA with Holm-Sidak's multiple comparisons test, $*p < 0.05$, $***p < 0.001$. Raptor, mTOR or S6K knockdown suppresses *Cd276* promoter activity in *Tsc2−/−* 105K cells (**c**) and *Tsc2* KO MEFs (**d**). $n = 6$, means ± SD, two-way ANOVA with Holm-Sidak's multiple comparisons test, $*p < 0.05$, $**p < 0.01$, $****p < 0.0001$. siRNA knockdown of YY2 reduces B7-H3 protein expression in *Tsc2−/−* 105K cells (**e**) and *Tsc2* KO MEFs (**f**) ($n = 3$).

Knockdown of YY2 reduces *Cd276* mRNA expression in *Tsc2−/−* 105K cells (**g**) and *Tsc2* KO MEFs (**h**). $n = 3$, means ± SD, two-tailed unpaired Student's t-test, $*p < 0.05$, $***p < 0.001$. **i** Promoter region of *Cd276* displaying the location of the YY2 binding site. YY2 ChIP-qPCR analysis showing increased YY2 occupancy on the *Cd276* promoter in *Tsc2−/−* 105K cells (**j**) and *Tsc2* KO MEFs (**k**). $n = 4$, means ± SEM, two-way ANOVA with Holm-Sidak's multiple comparisons test, $*p < 0.05$, $**p < 0.01$, $****p < 0.0001$. *Cd276* promoter activity is suppressed by YY2 knockdown in *Tsc2−/−* 105K cells (**l**) and *Tsc2* KO MEFs (**m**). $n = 6$, means ± SD, two-way ANOVA with Holm-Sidak's multiple comparisons test, $*p < 0.05$, $***p < 0.001$. Source data and exact $p$ values are provided in the Source data file.

To understand how S6K upregulates YY2, we first confirmed that wild-type S6K, but not kinase-dead (KD) S6K, increases the levels of YY2 in HeLa cells (Fig. 4e). To determine whether S6K and YY2 form a complex in the nucleus, we used immunofluorescence, and found that a fraction of exogenous and endogenous YY2 co-localizes with S6K in the nucleus of HeLa cells (Supplementary Fig. 2g). To confirm this result, we performed proximity ligation assay (PLA) and found complexes containing YY2 and S6K in the nucleus of HeLa cells (Supplementary Fig. 2h). On average, 5.5 puncta representing S6K/YY2 interaction were observed per nucleus ($n = 30$ cells) in HeLa cells versus 0.63 (S6K antibody plus IgG) and 0.47 (YY2 antibody plus IgG) puncta/nucleus in controls (Supplementary Fig. 2h).

To determine whether S6K directly phosphorylates YY2, we performed in vitro kinase assays in a cell-free system, using eIF4B, a known S6K substrate, as a positive control, and found that phospho-motif antibody (RXXS/T) recognizes YY2 when incubated with active S6K (Fig. 4f). Consistent with these data, inhibition of S6K1 activity in intact HeLa cells with PF-4708671 for 1 hour markedly suppressed YY2 phosphorylation at the RXXS/T site (Fig. 4g). This 1-hour timepoint was selected because it does not affect the total YY2 protein level, thereby facilitating the assessment of YY2 phosphorylation. Since the RXXS/T motif is also an Akt substrate motif, we immunoblotted for the phosphorylated form of Akt. We did not find a change in Akt phosphorylation (S473 and T308) (Fig. 4g). Interestingly, we found that PF-4708671 increases the poly-ubiquitination of YY2 (Fig. 4h), suggesting that phosphorylation of YY2 by S6K may inhibit its protein degradation. The YY2 protein sequence contains an evolutionarily conserved RXXT site (Fig. 4i). Mutating T336 to alanine (T336A) prevented recognition of the RXXT site by the RXXS/T antibody (Fig. 4j) and increased poly-ubiquitination, via an S6K1-dependent mechanism (Fig. 4k).

To identify the putative E3 ubiquitin ligase that mediates the ubiquitination of YY2, we used the publicly available database UbiBrowser[42] and identified Smurf1 as an unconfirmed E3 ubiquitin ligase for YY2 (Fig. 4l,m). Overexpression of Smurf1 suppressed YY2 protein levels (Fig. 4n) and increased YY2 poly-ubiquitination, which was rescued by the expression of WT-S6K (Fig. 4o). Finally, we found that Smurf1 binds to YY2 (Fig. 4p). To further confirm these results, we overexpressed Myc-YY2 in Tsc2-deficient 105K cells and examined B7-H3 expression after 24 hours of rapamycin treatment and found that YY2 overexpression restores B7-H3 expression in the rapamycin-treated cells (Fig. 4q). Together, these results indicate that S6K phosphorylates YY2 at T336, thereby inhibiting its ubiquitin-mediated degradation by Smurf1.

### Inhibition of B7-H3 promotes antitumor immunity in Tsc2-deficient tumors

Because B7-H3, an immune checkpoint protein, is markedly enriched in TSC-associated tumors, we sought to determine the impact of B7-H3 on the growth of TSC2-deficient tumors using our previously established models in which Tsc2-deficient cells (105K and TTJ) are subcutaneously (s.c.) inoculated into syngeneic mice[43]. We stably knocked down B7-H3 in 105K (Supplementary Fig. 3a) and TTJ cells

(Supplementary Fig. 3b). In vitro, no effect of B7-H3 knockdown on cell proliferation was observed in 2-D assays (Supplementary Fig. 3c, d) or in 3-D assays of anchorage-independent growth (Supplementary Fig. 3e, f). In vivo, the results were strikingly different: after s.c. inoculation of B7-H3 knockdown or control 105K cells into wild-type (WT) syngeneic C57BL/6 J mice, B7-H3 inhibition suppressed tumor growth by at least 80% ($p < 0.0001$) (Fig. 5a, b) and markedly increased the tumor-free survival ($p < 0.0001$) (Fig. 5c). To further confirm these results, CRISPR/Cas9 was used to knockout the *Cd276* gene in 105K cells (Supplementary Fig. 3g). These cells also exhibited slower growth in vivo (Supplementary Fig. 3h).

To examine the potential therapeutic benefit of B7-H3 inhibition, we treated mice carrying s.c. Tsc2-deficient 105K cells with neutralizing anti-B7-H3 antibody (Supplementary Fig. 3i). Mice that received B7-H3 antibody had smaller tumors (Supplementary Fig. 3j) with increased CD4+ T-cell (Supplementary Fig. 3k) and CD8+ T-cell (Supplementary Fig. 3l) infiltration. In parallel, we tested the efficacy of B7-H3 antibody treatment in *Tsc2+/−* mice, which develop renal cysts and tumors (cystadenomas)[44]. *Tsc2+/−* mice (7-month-old) were treated with isotype control antibody or anti-B7-H3 antibody for 24 days (Supplementary Fig. 3m). Anti-B7-H3 antibody reduced the burden of kidney tumors (Supplementary Fig. 3n, o), with the gross kidney tumor score decreased by 50% ($p < 0.001$) (Supplementary Fig. 3p) and the microscopic kidney tumor score reduced by 70% ($p < 0.001$) (Supplementary Fig. 3q). These results support the hypothesis that B7-H3 can be targeted for the treatment of tumors with hyperactivation of mTORC1.

Similar results were seen in Tsc2-deficient TTJ cells, with B7-H3 inhibition suppressing tumor growth by >50% (Supplementary Fig. 4a) and approximately doubling tumor-free survival (Supplementary Fig. 4b). After intravenous (IV) inoculation (used as a model of metastasis), B7-H3 knockdown reduced the number of lung nodules per lung by > 90% for 105K cells and >80% for TTJ cells (Supplementary Fig. 4c–f). Collectively, these data demonstrate a critical role of B7-H3 in the growth of mTORC1-hyperactive tumors.

These differences between the in vitro and in vivo impact of B7-H3 knockdown on tumor cell growth suggest that an intact immune system is required. To examine this, we injected B7-H3 knockdown and control Tsc2-deficient 105K cells into wild-type and *Rag1−/−* mice, which have a combined T and B cell immunodeficiency. In contrast to the results in immunocompetent mice in which B7-H3 knockdown strongly suppresses tumor cell growth, B7-H3 knockdown and control cells have similar tumor growth kinetics in *Rag1−/−* mice (Fig. 5d, e), thus implicating adaptive immunity as the main mechanism of B7-H3-mediated effects on tumor growth in cells with hyperactive mTORC1.

Having found that B7-H3 knockdown does not affect Tsc2-deficient tumor growth in immunodeficient mice (Fig. 5d, e), we next sought to define the infiltrating immune cells that are regulated by B7-H3 using Cytometry by time-of-flight (CyTOF). Cluster analyses were first visualized with vi-SNE plots on pre-gated single live CD45+ cells. Increased CD8+ and CD4+ T cells were observed in B7-H3 knockdown infiltrating tumors (Fig. 5f), with B7-H3 knockdown tumors contained

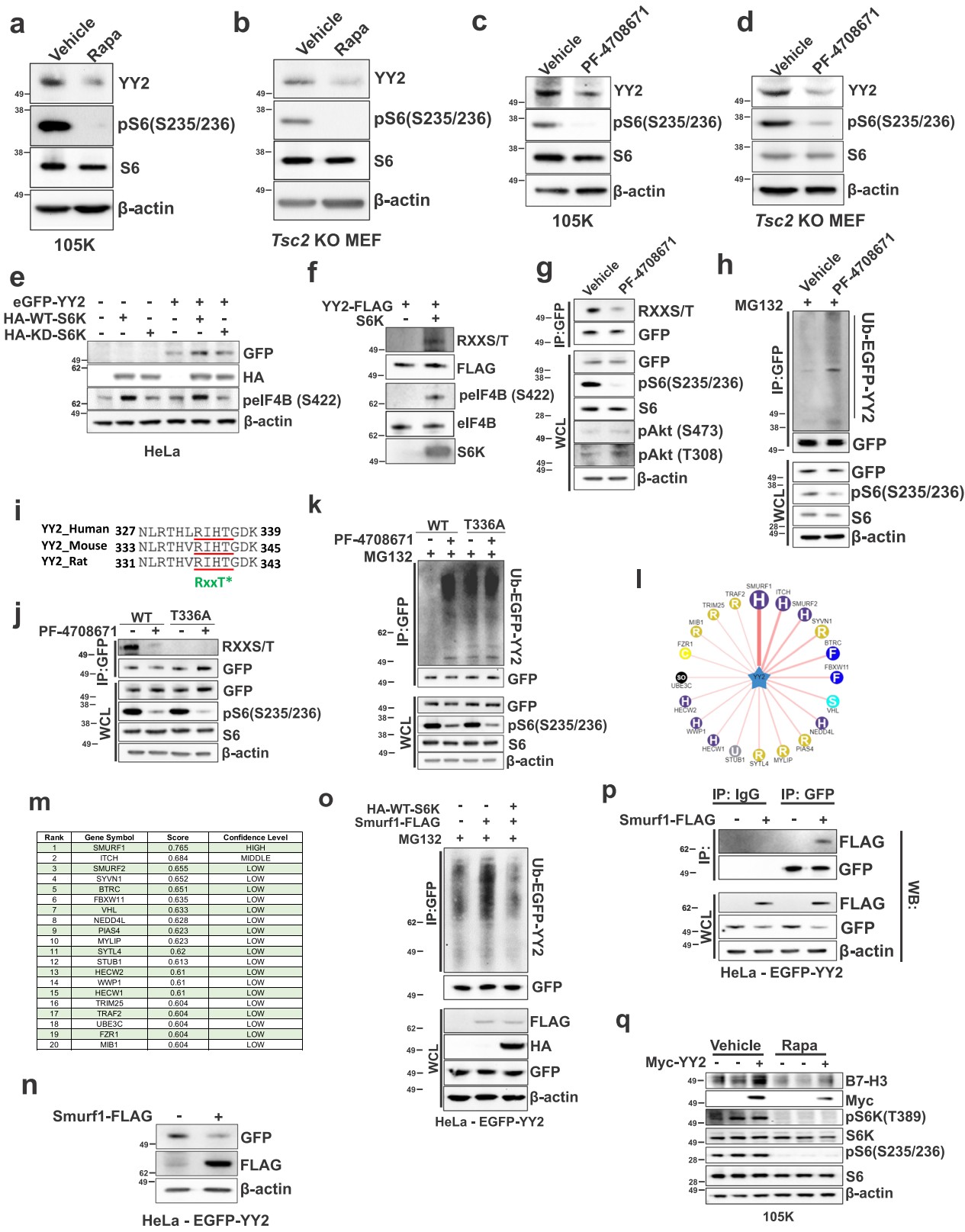

2-fold more CD4[+] T cells ($p < 0.05$) and 3-fold more CD8[+] T cells ($p < 0.05$) compared to controls (Fig. 5g). CD11b[+]Ly6G[+] polymorphonuclear myeloid-derived suppressor cells (PMN-MDSCs) were 80% lower in B7-H3 knockdown tumors (Fig. 5g). No apparent difference in F4/80[+], CD11c[+], CD19[+] or Ly6C[+] cells was observed (Fig. 5f, g). The CyTOF analyses were confirmed by immunohistochemical staining for CD3, CD8, and CD4 (Fig. 5h, i). Similar results were found in TTJ cells

with B7-H3 inhibition, with increased CD4[+] and CD8[+] T cells, and decreased PMN-MDSCs (Supplementary Fig. 4g–i). Immunohistochemistry confirmed that B7-H3 knockdown increased CD4[+] and CD8[+] T-cell infiltration in TTJ tumors (Supplementary Fig. 4j–l). It was worth noting that no differences in F4/80[+] macrophages, CD31[+] endothelial cells, or phosphorylated S6-positive cells were found in the B7-H3 knockdown tumors (Supplementary Fig. 4m).

**Fig. 4 | YY2 is a downstream target of the mTORC1/S6K signaling axis and is ubiquitinated by Smurf1.** YY2 protein expression in *Tsc2−/−* 105K cells (**a**) and *Tsc2* KO MEFs (**b**) treated with 20 nM rapamycin (Rapa) or vehicle for 24 hr (*n* = 3). YY2 protein expression in *Tsc2−/−* 105K cells (**c**) and *Tsc2* KO MEFs (**d**) treated with 10 μM PF-4708671 (S6K1 inhibitor) or vehicle for 24 hr (*n* = 3). **e** Immunoblot analysis of HeLa cells transfected with eGFP-YY2, wild-type S6K (HA-WT-S6K), or kinase-dead S6K (HA-KD-S6K) for 48 hr (*n* = 3). **f** Immunoblot analysis of in vitro kinase assays of recombinant active S6K in the presence or absence of myc-YY2 as a substrate. eIF4B was used as a positive control (*n* = 3). **g** Immunoblot analysis of immunoprecipitated (IP) GFP-YY2 and the whole-cell-lysate (WCL) from HeLa cells expressing GFP-YY2 (*n* = 3). **h** Immunoblot analysis of immunoprecipitated GFP-YY2 and the WCL from HeLa cells expressing GFP-YY2 (*n* = 3). **i** The evolutionarily conserved putative S6K phosphorylation site in YY2. **j** Immunoblot analysis of immunoprecipitated GFP-YY2 and the WCL from HeLa cells stably expressing wild-type (WT) GFP-YY2 or mutant (T336A) GFP-YY2, treated with 40 μM PF-4708671 for 1 hr (*n* = 3). **k** Immunoblot analysis of immunoprecipitated GFP-YY2 and the WCL from HeLa cells expressing WT GFP-YY2 or T336A GFP-YY2 (*n* = 3). **l, m** Network view of predicted E3 ubiquitin ligase-YY2 interactions by UbiBrowser 2.0. Confidence level of (**m**). **n** Decreased EGFP-YY2 expression in HeLa cells transfected with Smurf1-FLAG (*n* = 3). **o** Decreased Smurf1 ubiquitination in HeLa cells transfected with S6K. Immunoblot analysis of GFP-immunoprecipitates from EGFP-YY2 HeLa cells transfected with Smurf1-FLAG or HA-WT-S6K for 24 hr (*n* = 3). **p** Immunoblot analysis of GFP-and IgG control immunoprecipitates from EGFP-YY2 HeLa cells transfected with Smurf1-FLAG (*n* = 3). **q** Immunoblot analysis of B7-H3 expression in *Tsc2−/−* 105K cells, transfected with control or myc-YY2 for 48 hr and treated with 20 nM rapamycin or vehicle for the last 24 hr (*n* = 3). Source data is provided in the Source data file.

## Intact host IFN-γ, tumor cell IFN-γ signaling, and MHC-II expression are necessary to achieve a positive response to B7-H3 inhibition

Having discovered that blocking B7-H3 can inhibit Tsc2-deficient cell growth in vivo in the presence of an intact adaptive immune system, we sought to define the molecular mechanisms underlying this response. Tumor cells from control or B7-H3 knockdown Tsc2-deficient 105K tumors were isolated using the MACS Tumor Cell Isolation Kit and the transcriptomic profiles of these cells was analyzed by RNA-sequencing. We performed unbiased Gene Set Enrichment Analysis (GSEA) to identify candidate regulatory pathways. The interferon gamma (IFN-γ) response signature emerged as the second most upregulated pathway in B7-H3 knockdown cells when compared with control cells (Fig. 6a and Supplementary Fig. 5a). We confirmed this upregulation of IFN-γ response gene signature using a previously defined, separate set of IFN-γ pathway genes[45] (Fig. 6b). IFN-γ is known to enhance the expression of surface MHC-I[46], MHC-II[47], the processing and presentation of tumor-specific antigens[48], and activate the transcription factor STAT1[49]. To determine how MHC gene expression is impacted by B7-H3 inhibition, we used a list of MHC genes curated from the literature and discovered that multiple MHC-II genes are significantly increased in sorted B7-H3 knockdown tumor cells (Fig. 6c). Immunoblot analysis revealed increased MHC-II protein as well as increased phosphorylated STAT1 and total STAT1 in B7-H3 knockdown tumors (Fig. 6d), consistent with our RNA-sequencing data. Immunofluorescence analyses confirmed a striking increase in the nuclear localization of STAT1 (the activated form) and increased MHC-II expression in B7-H3 knockdown tumor cells when compared with control tumor cells (Fig. 6e, f and Supplementary Fig. 4n). It is worth noting that B7-H3 deficiency did not mitigate the differences in pSTAT1 and STAT1 levels in cultured cells in response to IFN-γ stimulation (Supplementary Fig. 5b, c). In addition, we did not observe a difference when comparing pSTAT1 and STAT1 levels in B7-H3 proficient and deficient cells in the presence of anti-CD3/CD28 beads-activated T cells isolated from the spleen of naïve mice (Supplementary Fig. 5d), implying that the tumor microenvironment is required for the enhanced IFN-γ responses in tumor cells with B7-H3 inhibition.

Interestingly, we found that Tsc2-deficient 105K, TTJ, and 621-101 cells, which do not express MHC-II/HLA-DR proteins under basal conditions, express low levels of MHC-II following 24 hours of IFN-γ stimulation and higher levels at 48 hours (Supplementary Fig. 5e–g). B7-H3 deficiency in cultured Tsc2-deficient cells does not affect IFN-γ-induced MHC-II expression (Supplementary Fig. 5h, i). The class II transactivator (CIITA) is a transcription factor that induces de novo synthesis of MHC-II[50]. We found that IFN-γ induces the expression of CIITA in Tsc2-deficient cells (Supplementary Fig. 5j–l) indicating that CIITA may be responsible for MHC-II expression in response to IFN-γ in the Tsc2-null cells. Next, we examined whether CIITA is responsible for

tumor alleviation mediated by B7-H3 inhibition. We used CRISPR-Cas9 to suppress CIITA expression, resulting in reduced MHC-II expression in B7-H3 knockdown 105K cells (Fig. 6g). Loss of CIITA increased the growth of B7-H3 knockdown 105K cells in vivo (Fig. 6h). These observations further underscore that the presence of an intact immune microenvironment is necessary for IFN-γ responses and MHC-II expression in B7-H3 knockdown tumors.

We next asked if host IFN-γ is actively involved in the tumor diminution mediated by B7-H3 deficiency. We compared the ability of Tsc2-deficient 105K cells with B7-H3 downregulation to grow in IFN-γ wild-type mice vs. strain-matched *IFN-γ−/−* mice. The B7-H3 shRNA tumor size was ~20-fold larger in *IFN-γ−/−* mice compared to wild-type mice at 30 days after inoculation (Fig. 6i, compare black and pink lines), suggesting host IFN-γ is necessary for the suppression of tumor growth. We then focused on T cells, which are the main source of IFN-γ in tumor microenvironment. Strikingly, we found increased IFN-γ+ and double-positive IFN-γ+ and TNF-α+ tumor-infiltrating CD8+ and CD4+ T cells in tumors with B7-H3 knockdown compared with controls (Fig. 6j, k). Direct co-culture of B7-H3 knockdown 105K cells with CD8+ T cells or CD4+ T cells significantly enhanced IFN-γ+ but not double-positive IFN-γ+ and TNF-α+ cells (Fig. 6l, m). This upregulation of IFN-γ was not observed when T cells were cultured with conditioned media from B7-H3 knockdown cells (Supplementary Fig. 5m, n). These data indicate that B7-H3 deficiency in the tumor cells enhances IFN-γ production in both CD8+ and CD4+ T cells and boost their ant-tumor properties in a cell-cell contact manner.

To further examine the impact of IFN-γ signaling in tumor cells, we used CRISPR-Cas9 to knockout the interferon gamma receptor 1 (IFNGR1) in B7-H3 knockdown 105K cells (Fig. 6n). Loss of IFNGR1 increased the growth of B7-H3 knockdown 105K cells in vivo (Fig. 6o), consistent with our prior data indicating that intact IFN-γ signaling is critical for the tumor-suppressive effects of B7-H3 inhibition.

## Intratumoral CD4+ T cells are required for the impact of B7-H3 knockdown on tumor growth

To determine the extent to which the antitumor effects of B7-H3 inhibition are dependent on CD8+ T cells vs. CD4+ T cells, CD8+ T cells or CD4+ T cells were depleted using anti-CD8α or anti-CD4 antibodies, respectively, in mice bearing 105K tumors with knockdown of B7-H3 using two different shRNAs or control shRNA (Fig. 7a–c and Supplementary Fig. 6a, b). Depleting either CD8+ T cells or CD4+ T cells significantly increased tumor volume in the cells with B7-H3 shRNAs (Fig. 7b, c and Supplementary Fig. 6a, b), but to our surprise, depletion of CD4+ T cells increased tumor growth more than depletion of CD8+ T cells (Fig. 7b, c), despite similar efficacy of depletion (>99% in both spleens and tumors) (Supplementary Fig. 6c, d). This difference was not observed in mice bearing shRNA control tumors as depletion of either CD4+ T cells or CD8+ T cells led to a similar increase in tumor volume (Supplementary Fig. 6a, b). Consistent with these results, B7-

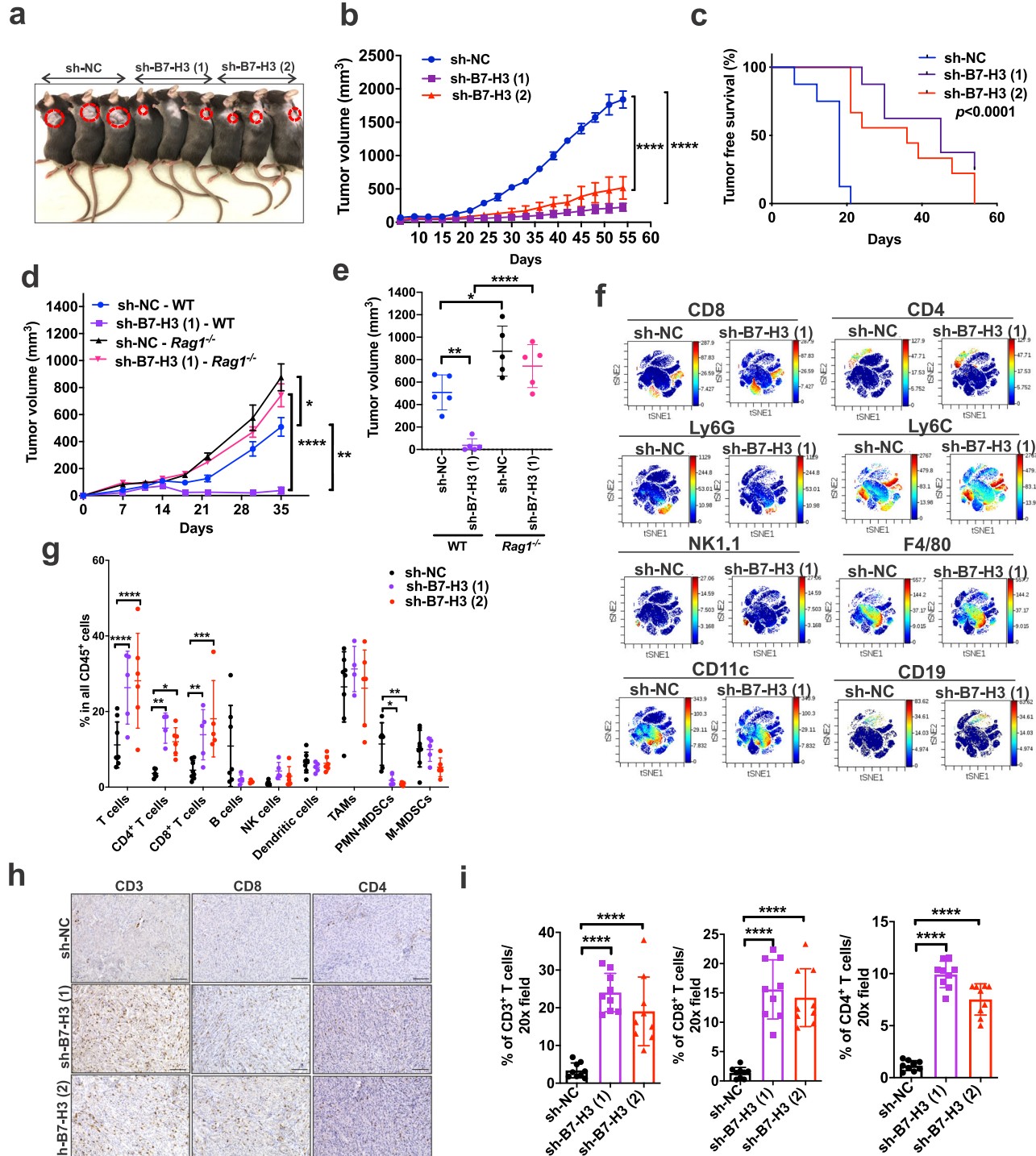

**Fig. 5 | Suppression of Tsc2-null tumor growth by inhibiting B7-H3 requires an intact adaptive immune system. a** Knockdown of B7-H3 using two different shRNAs in *Tsc2−/−* 105K cells inhibits subcutaneous tumor growth in syngeneic wild-type (WT) C57BL/6 J mice compared with non-targeting shRNA (sh-NC). **b** Growth of subcutaneous sh-NC, sh-B7-H3 (1), and sh-B7-H3 (2) *Tsc2−/−* 105K tumors in WT C57BL/6 J mice. *n* = 8 mice/group, means ± SD, one-way ANOVA with Holm-Sidak's multiple comparisons test, ****$p < 0.0001$. **c** Tumor-free survival curve of mice in **b**. Log-rank analysis. **d** Growth of subcutaneous sh-NC or sh-B7-H3 (1) *Tsc2−/−* 105K tumors in WT or *Rag1−/−* C57BL/6 J mice. *n* = 5 mice/group, means ± SEM, two-way ANOVA with Holm-Sidak's multiple comparisons test, *$p < 0.05$, **$p < 0.01$ ****$p < 0.0001$. **e** Tumor volume of **d** 35-day post cell inoculation. *n* = 5 mice/group, means ± SD, two-way ANOVA with Holm-Sidak's multiple comparisons

test, *$p < 0.05$, **$p < 0.01$ ****$p < 0.0001$. **f** Density viSNE plots from CyTOF data on pre-gated CD45+ tumor-infiltrating lymphocytes (TILs) from sh-NC or sh-B7-H3 (1) *Tsc2−/−* 105K tumors. **g** Percentage of indicated cell types within CD45+ TILs in **f**. *n* = 6 or 8 tumors for sh-NC group, *n* = 4 or 5 tumors for sh-B7-H3 (1), *n* = 4, 5, or 6 tumors for sh-B7-H3 (2). Exact *n* is indicated in Source data file. Means ± SD, two-way ANOVA with Holm-Sidak's multiple comparisons test, *$p < 0.05$, **$p < 0.01$, ***$p < 0.001$, ****$p < 0.0001$. **h** Top panel: Representative images of CD3, CD4 and CD8 IHC staining on tumor sections from each group. Scale bar = 100 μm. **i** Quantification of CD3+, CD4+ and CD8+ T cells in each group (*n* = 9/group). Means ± SD, one-way ANOVA with Dunnett's multiple comparisons test, ****$p < 0.0001$. Source data and exact *p* values are provided in the Source data file.

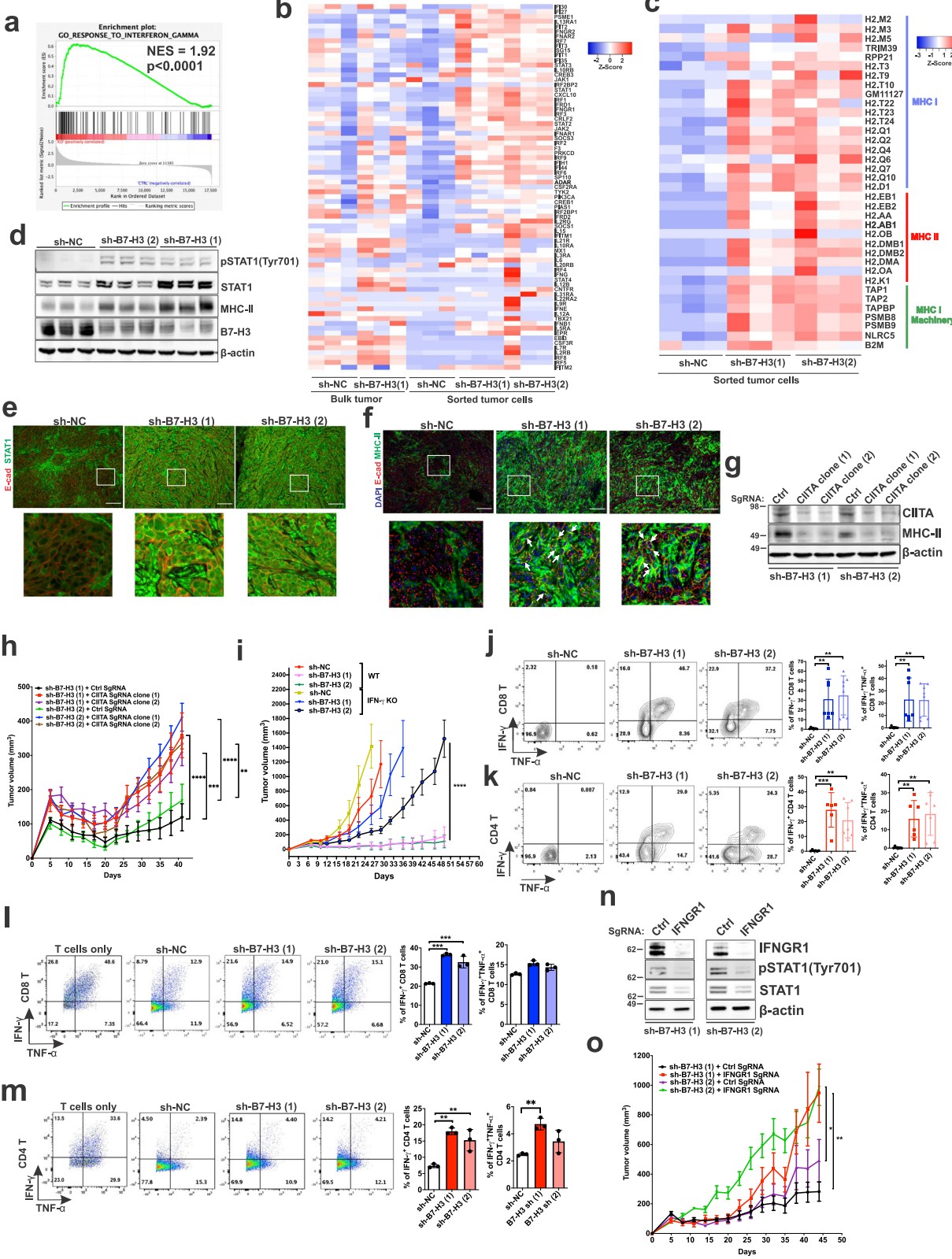

H3 knockdown tumors grew faster in *Cd4* KO mice than *Cd8* KO mice (Supplementary Fig. 6e), with tumor size ~4-fold larger in *Cd4* KO mice vs. control mice, and ~2.7-fold larger in *Cd8* KO mice (Supplementary Fig. 6f), whereas the growth of the shRNA control cells was not different between *Cd4* KO mice and *Cd8* KO mice (Supplementary Fig. 6g, h). Together, these data suggest that CD4+ T cells play an important role in controlling the growth of B7-H3-deficient tumors.

## B7-H3 inhibition increases intratumoral cytolytic CD38+CD39+CD4+ T cells

Next, we performed Cellular Indexing of Transcriptomes and Epitopes (CITE-seq) to investigate how B7-H3 regulates the dynamics of tumor-reactive T cells in the microenvironment. We FACS-sorted total live intratumoral CD3+ T cells from Tsc2-deficient 105K tumors with either B7-H3 knockdown or control shRNA (80 tumors from each group were

**Fig. 6 | Inhibition of B7-H3 increases expression of IFN-γ signature genes and MHC-II genes via enhancement of effector CD4+ and CD8+ T cell responses.**
**a** IFN-γ is one of the top activated pathways in B7-H3 knockdown tumors versus control tumors ($n = 3$/group). NES = normalized enrichment score. **b** Heatmap of IFN-γ-induced gene expression in bulk tumors and sorted tumor cells of the indicated groups ($n = 3$/group). **c** Heatmap of unsupervised clustering of MHC-I/II gene expression in sorted tumor cells from each group ($n = 3$/group). **d** Immunoblot analysis of tumor lysates derived from the indicated tumors. ($n = 3$/group). **e** Representative images of E-cadherin (E-cad) and STAT1 immunofluorescent staining of tumor sections ($n = 3$/group). Scale bar = 100 μm. **f** Representative images of E-cad and MHC-II immunofluorescent staining on tumor sections ($n = 3$/group). White arrows indicate co-expression in the same cells. Scale bar = 100 μm. **g** Immunoblot analysis of Tsc2−/− 105K cells stably expressing sh-B7-H3 (1) or sh-B7-H3 (2) and control (Ctrl) or CIITA sgRNAs ($n = 3$). **h** CIITA knockout increases the subcutaneous growth of tumor cells in (**g**) in mice. $n = 10$ mice/group, means ± SEM, two-way ANOVA with Holm-Sidak's multiple comparisons test, **$p < 0.01$,

***$p < 0.001$, ****$p < 0.0001$. **i** Growth of subcutaneous sh-NC, sh-B7-H3 (1), or sh-B7-H3 (2) Tsc2−/− 105K tumors in strain-matched WT or IFN-γ KO mice. $n = 10$ mice/group, means ± SD, Mann-Whitney U Test, ****$p < 0.0001$. Percentage of IFN-γ/TNF-α positive tumor-infiltrating CD8+ (**j**) and CD4+ (**k**) T cells in the indicated tumors. $n = 6, 7$ or 8/group. Exact $n$ is indicated in Source data file. Means ± SD, one-way ANOVA with Dunnett's multiple comparisons test, **$p < 0.01$, ***$p < 0.001$. Percentage of IFN-γ/TNF-α in WT splenic CD8+ (**l**) and CD4+ (**m**) T cells after co-culture with Tsc2−/− control, B7-H3 (1) or B7-H3 (2) shRNA 105K cells. $n = 3$, means ± SD, one-way ANOVA with Dunnett's multiple comparisons test, **$p < 0.01$, ***$p < 0.001$.
**n** Immunoblot analysis of Tsc2−/− 105K cells stably expressing B7-H3 (1) or B7-H3 (2) shRNA and control (Ctrl) or IFNGR1 sgRNAs ($n = 3$). **o** IFNGR1 knockout increases the subcutaneous growth of tumor cells in (**n**) in mice. $n = 10$ mice/group, means ± SEM, two-way ANOVA with Holm-Sidak's multiple comparisons test, *$p < 0.05$, **$p < 0.01$. Source data and exact $p$ values are provided in the Source data file.

pooled). The isolated CD3+ T cells were incubated with a panel of 190 barcoded antibodies, including isotype controls, and then captured as individual cells using a droplet-based scRNAseq platform (Chromium, 10X Genomics). After sequencing and demultiplexing, we performed rigorous quality control to remove doublets and to generate a total of 12,820 high-quality single-cell transcriptomes (7,885 cells from shRNA control tumors and 4,935 from B7-H3 knockdown tumors). Data were integrated using the reciprocal PCA (RPCA) algorithm[51]. CD4+ and CD8+ T cells could be identified by mRNA and protein expression based on a UMAP generated from the top 30 principal components (Fig. 7d, e). The CD4+ T cell population was increased in B7-H3 knockdown tumors compared with control tumors (40% of cells versus 25%) (Fig. 7f). CD4+ T cells from B7-H3 knockdown tumors exhibited elevated expression of genes linked to effector/cytolytic functions, including *Gzmk*, *Prf1*, *Nkg7*, and *Cst7*, as well as markers of exhaustion (for example, *Pdcd1* and *Lag3*) (Fig. 7g). We also observed increased expression of effector/cytokines, including *Ifng* and *Tnf*, suggesting that these CD4+ T-cell populations display a more activated than exhausted T-cell profile (Fig. 7g). GSEA revealed that CD4+ T cell populations from B7-H3 knockdown tumors are enriched for genes involved in NK-like cytotoxicity and PI3K/Akt signaling (Fig. 7h). We found similar elevated expression of genes involved in effector/cytolytic/cytokine function in CD8+ T cells, including *Gzmk*, *Gzmb*, *Prf1*, *Ifng*, and *Nkg7* (Fig. 7i). GSEA of CD8+ T cells from B7-H3 knockdown tumors showed enrichment in ongoing T cell activation, proliferation, response to IFN-γ, and NK-like cytotoxicity (Fig. 7j).

Using unsupervised clustering and differential gene expression, we resolved the intratumoral T cell subsets and functional states into 10 refined clusters. Clusters were annotated based on the expression of known genes. CD4+ and CD8+ T cells each comprised 5 clusters; resting cells expressing *Sell*, *Tcf7* and *Ly6c1* were defined as naïve-like CD4+ or CD8+ T cells (Fig. 7k, l). Two activation-effector clusters were identified in CD4+ T cells: one cluster showed high expression of *Il18r1* as well as *Cd44*, *Nrp1*, *Tbc1d4* and was referred to as Il18r1 CD4+ T cells; the other cluster had high expression of *Gzmk*, *Pdcd1*, *Cdk6*, *Ifng*, *Lag3*, *Ccl3*, and *Ccl4* and was annotated as CD4_Teff-Gzmk (Fig. 7k, l). High expression of *Gzmk* and *Gzmb* was found in CD8+ effector T cells and referred to as CD8_Teff-Gzmk/Gzmb. We also found two clusters of Tregs based on the expression of *Foxp3* and *Il2ra*; one had high and the other had low expression of *Pdcd1* and *Klrg1*, referred to as Tregs-PD1/KLRG1 and Tregs, respectively (Fig. 7k, l). Of note, we also found Mki67-proliferating cells (CD8_Tprof-Mki67), early activation cells (T_early_act), and *Ly6c2* high cells (CD8_T-Ly6C) (Fig. 7k, l). Importantly, the percentage of CD4_Teff-Gzmk and CD8_Teff-Gzmk/Gzmb was increased by 6-fold and 2.5-fold in B7-H3 knockdown tumors compared to control, respectively (Fig. 7m).

To screen for potential surface markers of effector/cytolytic CD4+ T cells, we exploited the cell surface protein information from the

CD4_Teff-Gzmk subcluster. We identified CD39 and CD38 as the two predominantly expressed cell surface proteins in the CD4_Teff-Gzmk subcluster (Fig. 7n). In addition to CD38 and CD39, but to a lower extent, CD4_Teff-Gzmk expressed CD200R, CD43, CD54, and PD-1 (Fig. 7n). From our CITE-seq data, we found that CD38 and CD39 are co-expressed on the same CD4+ TILs, and that CD39 has the highest Pearson correlation score with CD38 (Supplementary Fig. 7a, b). We validated that the number of CD38+CD39+CD4+ TILs was elevated in B7-H3 knockdown tumors by immunofluorescent staining (Fig. 7o). We also confirmed that the elevated CD38+CD39+CD4+ TILs are not Tregs by excluding CD25-positive CD4+ T cells using flow cytometry (Supplementary Fig. 7c). Consistent with the data from the s.c. models, the number of CD38+CD39+CD4+ TILs in tumor foci in the lungs of mice injected with B7-H3 knockdown was increased compared to mice injected with control cells (Supplementary Fig. 7d). To verify the biological function of non-Treg CD38+CD39+CD4+ TILs, we isolated CD38+CD39+CD25-CD4+ TILs by FACS from Tsc2-deficient B7-H3 knockdown 105K tumors and cultured the cells ex vivo with interleukin-2 (IL-2). We co-cultured these cells with Tsc2-deficient 105K tumor cells and employed an imaging-based time-lapse cytolytic assay, assessing cell death with Cytotox Green. The CD38+CD39+CD25-CD4+ TILs triggered increased tumor cell death. Pre-treating tumor cells with MHC-II blocking antibody inhibited CD38+CD39+CD25-CD4+ TIL-mediated tumor cell death (Fig. 7p). These data indicate that CD38+CD39+CD25-CD4+ TILs not only express granzymes and perforin, but can also recognize tumor antigen in an MHC class II-dependent manner and are functionally capable of lysing tumor cells.

To determine the clinical relevance of the CD38+CD39+CD4+ T-cell signature identified in our studies in cancer patients, we used the top 17 upregulated genes identified in the CD38+CD39+CD4+ T effector_Gzmk cells from CITE-seq RNA data and performed gene set variation analysis of the entire TCGA dataset and generated a cytotoxic CD4 T-cell score for each patient. We found high cytotoxic CD4+ T score is associated with a better overall survival in cancer patients (Fig. 7q; $p = 0.0396$).

We also analyzed a published scRNA-seq dataset of 8 metastatic RCC patients before and after anti-PD-1 immune checkpoint blockade (ICB) for non-Treg CD38+CD39+CD4+ TILs[52]. We performed unsupervised clustering on the CD4+ T cells identified in this study. CD4+ T cells formed four distinct clusters, with the dominant population marked by expression of *GZMK*, *GZMB*, *CCL4* and *CCL5* (CD4-GZMK), consistent with the cytotoxic CD4+ TILs we identified in our CITE-seq data (Supplementary Fig. 7e, f). We also found three other populations differentiated by their expression of CP (CD4-CP), APOE (CD4-APOE) and FOXP3 (Tregs) (Supplementary Fig. 7e, f). The proportion of CD4-GZMK cells was higher in patients who responded to ICB than in patients who did not respond (Supplementary Fig. 7g, h). Most importantly, ICB-exposed responders had more CD4-GZMK cells expressing *CD38* and *ENTPD1* (gene name of CD39) (Supplementary Fig. 7i, j).

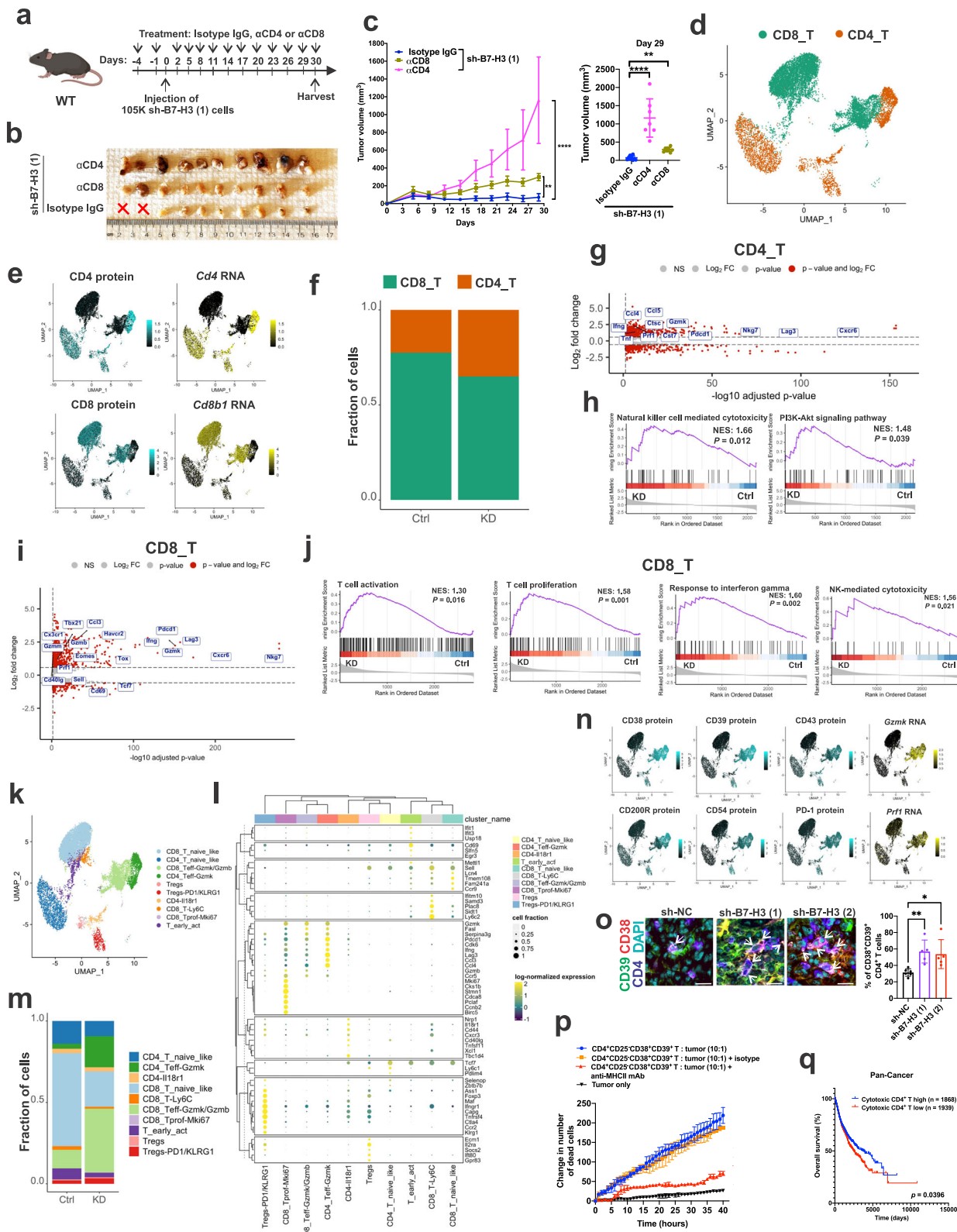

## Discussion

B7-H3 mRNA and protein are overexpressed in the majority of human cancers[53,54], and clinical trials of anti-B7-H3 antibody are underway for melanoma, squamous cell cancer of the head and neck (SCCHN), and non-small cell lung cancer (NSCLC). Here we show high B7-H3/*CD276* expression together with a high mTORC1 activity score is associated with poorer clinical outcomes and immunosuppressive features. The

underlying molecular mechanisms responsible for B7-H3 expression are not well-characterized. Tissue microarrays including multiple tumor types revealed that higher levels of phospho-S6 (a marker of mTORC1 activation) are associated with higher expression of B7-H3. In TSC2-deficient cells, we found that mTORC1 drives the expression of B7-H3 via the YY2 transcription factor, which binds the promoter of B7-H3 and drives B7-H3 transcription. We further demonstrate that S6K, a

**Fig. 7 | Suppression of B7-H3 increases intratumoral cytotoxic CD38⁺CD39⁺CD4⁺ T cells.** a Experimental design for B7-H3 knockdown *Tsc2–/–* 105K tumors treated with isotype control, α-CD4, or α-CD8 (Created with Biorender.com). **b** Images of B7-H3 knockdown *Tsc2–/–* 105K tumors treated as depicted in **a**. X indicates that no tumor was detected. *n* = 10 mice for isotype control, and α-CD8, and *n* = 7 mice for α-CD4. **c** Left panel: Suppressed growth of B7-H3 knockdown *Tsc2–/–* 105K tumors treated with anti-CD8 or anti-CD4 antibodies. Right panel: Tumor volume at 29 days post-cell injection. *n* = 10 mice for isotype control, and α-CD8, and *n* = 7 mice for α-CD4. Means ± SD, nonparametric Kruskal-Wallis test with Dunn's multiple comparisons test, **\**p* < 0.01, \*\*\*\**p* < 0.0001. **d** Uniform manifold approximation and projection (UMAP) plots of 12,820 sorted single CD3⁺ T cells from sh-NC tumors (*n* = 80 tumors pooled into 2 independent samples) or sh-B7-H3 (1) tumors (*n* = 80 tumors pooled into 2 independent samples). CD4⁺ and CD8⁺ T cells are indicated. **e** Expression of CD4 and CD8 based on CITE-seq antibody binding and RNA-seq. **f** Percentage of tumor-infiltrating CD4⁺ and CD8⁺ T-cells as a proportion of total CD3 T cells in *Tsc2–/–* 105K control (Ctrl) or B7-H3 knockdown (KD) tumors. *n* = 80 tumors pooled into 2 independent samples/group. **g** Volcano plot depicting differentially expressed genes in tumor-infiltrating CD4⁺ T cells from *Tsc2–/–* 105K B7-H3 KD versus Ctrl tumors of the CITE-seq RNA data. **h** GSEA of the CITE-seq RNA data from tumor-infiltrating CD4⁺ T-cells from *Tsc2–/–* 105K Ctrl and B7-H3 KD tumors of the CITE-seq RNA data. NES, normalized enrichment score. **i** Volcano plot depicting differentially expressed genes in tumor-infiltrating CD8⁺ T cells from *Tsc2–/–* 105K B7-H3 KD versus Ctrl tumors of the CITE-seq RNA data. **j** GSEA of the CITE-seq RNA data from tumor-infiltrating CD8⁺ T cells from *Tsc2–/–* 105K Ctrl and B7-H3 KD tumors. NES, normalized enrichment score. **k** UMAP showing minor phenotypic clusters of **d**. **l** Dotplot depicting the top 62 upregulated genes in the minor clusters of CD8⁺ and CD4⁺ T cells. **m** Percentage of tumor-infiltrating CD4⁺ and CD8⁺ T-cell minor clusters shown as a proportion of total CD3⁺ T cells isolated from *Tsc2–/–* 105K Ctrl or B7-H3 KD tumors. *n* = 80 tumors pooled into 2 independent samples/group. **n** Epitope and mRNA signals in cells from CITE-seq experiments for selected top upregulated markers in CD4_Teff-Gzmk and CD8_Teff-Gzmk/Gzmb clusters. **o** Representative images and quantification of CD38⁺CD39⁺CD4⁺ TILs in sh-NC, sh-B7-H3 (1), or sh-B7-H3 (2) *Tsc2–/–* 105K tumors. White arrows indicate CD38⁺CD39⁺CD4⁺ TILs. *n* = 6 tumors for sh-NC, and *n* = 5 tumors for sh-B7-H3 (1), and sh-B7-H3 (2). Mean ± SD, nonparametric Kruskal-Wallis test with Dunn's multiple comparisons test, \**p* < 0.05, \*\**p* < 0.01. Scale bar = 10 μm. **p** Quantification of apoptotic cells over time with *Tsc2–/–* 105K cells cultured alone or with tumor-infiltrating CD38⁺CD39⁺CD25⁻CD4⁺ T cells. Mean ± SD from 3 independent experiments. **q** Kaplan-Meier graph from the pan-TCGA data showing improved overall survival of patients with high cytotoxic CD4⁺ T-cell gene signature. Source data and exact *p* values are provided in the Source data file.

key substrate of mTORC1, directly phosphorylates YY2 at Threonine 336. Despite these and other data supporting a key role of B7-H3 in many types of cancer, the mechanisms through which B7-H3 promotes tumorigenesis are not fully understood. We found that inhibition of B7-H3 in mouse models with high mTORC1 activity inhibits tumor growth, associated with increased infiltration of cytotoxic CD38 and CD39-double-positive CD4⁺ T cells and increased IFN-γ responses and MHC-II expression. This work reveals fundamental mechanisms through which B7-H3 is regulated in cancer and broadens our understanding of the impact of the B7-H3 immune checkpoint molecule on cancer-immune cell communication.

YY2 is a paralog of YY1 in humans and mice. The two transcription factors have a high level of sequence identity, with ~56% and 65% in their overall amino acid sequences and DNA-binding motif, respectively[55]. Growing evidence indicates that YY2 is not redundant to YY1, since inhibition of either YY2 or YY1 results in inverse changes in ultraviolet sensitivity, cell proliferation and regulation of IL-4 promoter activity[56,57]. Additionally, YY1 and YY2 can regulate distinct gene sets[56]. Despite these differences, the vast majority of prior research has focused on YY1, and not YY2. We found that the protein level of YY2, but not YY1, is upregulated in TSC2-deficient cells, in which mTORC1 activity is high, and that inhibition of YY2, but not YY1, suppresses B7-H3 expression, indicating YY2 specific regulation. We also found that YY2 binds to the B7-H3 gene promoter region (Fig. 3j, k), demonstrating that YY2 is a key inducer of B7-H3 downstream of mTORC1.

The discovery that YY2 is upregulated at the protein but not mRNA level in TSC2-deficient cells with mTORC1 hyperactivity led us to investigate YY2's regulatory mechanisms, which are largely uncharacterized. Our studies reveal that YY2 is directly phosphorylated by S6K at the T336 residue in the C-terminus (Fig. 4j). This phosphorylation enhances YY2 protein expression by blocking its ubiquitin-mediated degradation via the E3 ubiquitin ligase Smurf1 which, as we demonstrate directly, binds and ubiquitinates YY2 (Fig. 4n, p). These findings add substantially to the limited prior evidence of YY2's regulation, which includes one prior study demonstrating that lysine residue 247 is methylated and demethylated by SET7/9 and LSD1, respectively, impacting its DNA-binding activity[58]. Interestingly, YY1's degradation is regulated by Smurf2, which is a closely related C2-WW-HECT domain E3 ubiquitin ligase of Smurf1[59].

B7-H3 downregulation results in markedly decreased tumor growth in mouse models with loss of TSC2 and mTORC1 hyper-activation (Fig. 5a–c and Supplementary Fig. 3g, h, 4a–f). The pro-tumorigenic mechanism of B7-H3 in Tsc2-null models appears to be distinct from prior models, which focused on CD8⁺ T cells and NK cells[10,11,60]. We found that growth is highly dependent on CD4⁺ T cells while earlier studies did not find an impact of CD4⁺ T cells[10,11]. Our data suggest that these differences could be related to MHC-II expression. In general, tumor cells widely express MHC-I molecules but do not express MHC-II molecules, partly due to defects in the production of the transcription factor CIITA[61], which controls the transcription of MHC-II[62]. We found that reduced B7-H3 inhibition upregulates MHC-II expression in Tsc2-deficient tumor cells in vivo, where high IFN-γ producing T cells are present in the tumor milieu (Fig. 6c, f, h, i). We speculate that enhanced MHC-II expression on B7-H3-deficient tumor cells together with professional antigen-presentation cells leads to a greater antigen presentation to cytolytic CD4⁺ T lymphocytes. Consistent with this hypothesis, some tumor cells, including multiple myeloma and some melanoma cell lines, demonstrate intrinsic or IFN-γ-stimulated expression of MHC-II[63–65], and the expression of MHC-II in melanoma correlates with CD4⁺ T-cell infiltration[64,65]. Our data expand the evolving understanding of the mechanisms through which B7-H3 upregulation promotes tumorigenesis.

Activation of the IFN-γ signaling pathway in tumor cells plays a key role in stimulating antitumor immune responses. Loss of genes in IFN-γ pathway serves as a resistance mechanism to anti-CTLA-4 blockade[45]. We found that the slow tumor growth kinetic resulting from B7-H3 inhibition is markedly dependent on the presence of host IFN-γ and activation of the IFN-γ:IFNGR1 pathway in tumor cells (Fig. 6g, n): while knockdown of B7-H3 decreased Tsc2-deficient tumor growth by ~90% in wild-type mice, it had little effect in IFN-γ knockout mice or when IFNGR1 expression is knocked down in the tumor cells. This suggests that tumors with an intact IFN-γ pathway are more likely to respond to therapy targeting B7-H3. We observed upregulation of PD-L1 and IDO transcripts from our B7-H3 knockdown tumor RNA-seq (Supplementary Data 1), which is of interest because both genes have been demonstrated to play a critical role in primary resistance to PD-L1/PD-1 immunotherapy[66,67]. In addition, MHC-II⁺ melanoma tumors have been shown to promote selective adaptive resistance to anti-PD-1 immunotherapy via upregulation of Lag-3 and FCRL6[64].

The relationship between B7-H3 expression and PD-L1 expression may be clinically relevant. Although, multiple data sets indicate that PD-L1 is not upregulated at baseline in TSC-associated tumors, including our prior work using immunohistochemistry[43] and our recent work using single cell RNA sequencing[68]. Interestingly, we examined PD-L1 (CD274) expression in our ex vivo sorted tumor cells and found that PD-L1 (CD274) is upregulated by 2.56-fold (padj = 1.46E-10) in sh-B7-H3 tumor cells compared to sh-NC (Supplementary Data 1). These data could provide a rationale for testing B7-H3

blockade in combination with PD-1/PD-L1 blockade therapy in tumors with mTORC1 hyperactivation.

It is known that antigen-specific cytotoxic CD4[+] T cells can be generated in the tumor milieu[69–71]. Our CITE-seq analyses reveal that B7-H3-depleted tumors have increased CD38 and CD39-double-positive CD4[+]CD25[-] T cells, and these cells possess cytolytic properties. Our discovery of CD38 and CD39 expression on the surface of cytotoxic tumor-specific CD4[+] T cells will allow the identification and isolation of this cell type for future studies and clinical trials relating to CD4[+] T-cell specific immunotherapies. Both CD38 and CD39 are ectoenzymes responsible for generating extracellular adenosine, which promotes an immunosuppressive tumor immune microenvironment[72,73]. Humanized anti-CD38 (Daratumumab), anti-CD39 (ES002023, NCT05075564) antibodies, and anti-CD38 CAR-T (NCT03464916) have entered or completed early phase clinical trials[74].

In summary, the data presented in this study demonstrate that mTORC1-hyperactive tumor cells escape immune-mediated killing via the induction of B7-H3 expression. This occurs due to direct phosphorylation of the YY2 transcription factor by S6K, thereby enhancing its stability and binding to the B7-H3 promoter. Inhibition of B7-H3 upregulates cytotoxic CD4[+]CD25[-]CD38[+]CD39[+] T cells leading to immune-mediated suppression of tumor progression and development. The effects of B7-H3 on tumor growth require host/tumor cell IFN-γ signaling as well as expression of MHC-II on tumor cells. These data reveal critical regulatory mechanisms within tumor cells that lead to B7-H3 upregulation and provide a detailed understanding of the consequences on the tumor immune microenvironment mediated by B7-H3. Lastly, from a translational perspective, these discoveries provide a mechanistic foundation for therapeutic development relating to B7-H3 and CD4[+] T cells.

## Methods

### Mice
C57BL/6 J, Ifngtm1Ts/J, Rag1tm1Mom/J, Cd4tm1Mak/J and Cd8atm1-Mak/J in C57BL/6 J background mice were purchased from Jackson Laboratory and Tsc2[+/-] mice in AJ background were kindly provided by the Tuberous Sclerosis Alliance Preclinical Research Consortium. Both female and male mice were used in this study.

### Cell lines
Tsc2[-/-] 105K renal tumor cells stably expressing empty vector or TSC2[75,76]; Tsc2 WT and Tsc2 KO MEFs;[77] 621-101 cells stably expressing empty vector or TSC2[78]; Tsc2[-/-] TTJ renal tumor cells[79] were cultured in DMEM (ThermoFisher Scientific #11995073), 10% fetal bovine serum (R&D SYSTEMS #S11150) and 1% penicillin-streptomycin (Sigma #P4333). Raptor inducible MEFs (iRapKO) and Rictor inducible MEFs (iRicKO)[80] were grown in DMEM, 10% fetal bovine serum, 1% penicillin-streptomycin and 2.5 μg of puromycin (ThermoFisher Scientific #A11138-03). Cancer cell lines (A549 (Cat#CCL-185), T47D (Cat#HTB-133) and PC3 (Cat#CRL-1435)) were from ATCC. Raptor and Rictor inducible MEFs were the kind gifts of Michael Hall.

### In vivo experiments in mouse tumor models
All animal procedures were approved by the Brigham and Women's Hospital Institutional Animal Care and Use Committee. The maximum tumor size (2 cm) in any direction permitted by ethics committee was not exceeded. Mice were housed in an animal facility with 12 h light/12 h dark cycle at 72 °F and 40% humidity with unrestricted access to food and water. For subcutaneous tumor inoculation, Tsc2[-/-] 105K or Tsc2[-/-] TTJ renal tumor cells (2.5 × 10[6]) stably expressing sh-NC (non-targeting control), sh-B7-H3 (1), sh-B7-H3 (2), sg-Ctrl, sg-CD276 (1), sg-CD276 (2), sh-B7-H3 (1) + Ctrl SgRNA, sh-B7-H3 (1) + CIITA SgRNA clone (1), sh-B7-H3 (1) + CIITA SgRNA clone (2), sh-B7-H3 (2) + Ctrl SgRNA, sh-B7-H3 (2) + CIITA SgRNA clone (1), sh-B7-H3 (2) + CIITA SgRNA clone (2), sh-B7-H3 (1) + INFGR1 SgRNA, or sh-B7-H3 (2) + INFGR1 SgRNA were

resuspended in 75 μl of growth factor free DMEM, and then 75 μl of growth factor reduced matrigel (Corning #356237) was added to the cell suspension. The cell/matrigel suspensions were injected subcutaneously into the flanks of six-week-old male C57BL/6 J mice (Jackson Laboratory). Tumor-free survival in tumor-bearing mice was defined as when tumors are not palpable and less than 100 mm[3].

For B7-H3 antibody treatment experiments, seven-month-old Tsc2[+/-] mice on the A/J strain background[81,82] and syngeneic subcutaneous Tsc2[-/-] 105K tumors were used[43]. Intraperitoneal injections of rat IgG1 isotype control antibody (BioxCell Cat#BE0088) or anti-B7-H3 (300 μg/mouse, BioxCell Cat#BE0124) antibody were administered to randomly assigned mice. Mice were treated every 3 days for 8 treatments in total. Mice were sacrificed 24 hr after the last treatment. For Tsc2[+/-] mice, the severity of renal kidney tumors was scored using previously established gross and microscopic kidney tumor scoring methods[83,84]. The gross kidney tumor score was determined as a summed score for all tumors in a kidney according to size: <1 mm, score 1; 1–1.5 mm, score 2; 1.5–2 mm, score 5; >2 mm, score 10. For the microscopic kidney tumor score, kidneys were first cut into 1 mm–thick pieces and then fixed in 10% formalin. Each tumor or cyst was measured (length, width), and the percent of the lumen filled by tumor was determined (0% for a simple cyst and 100% for a completely filled, solid tumor) from 5-μm H&E sections for each of the 1 mm–thick pieces for each kidney. These measurements were converted into a score as previously established[84]. For the subcutaneous model, Tsc2[-/-] 105K renal tumor cells (2.5 × 10[6]) injected subcutaneously into six-week-old male C57BL/6 J mice were allowed to grow to 150-200 mm[3] before initiation of B7-H3 antibody therapy.

### In vivo experiments in mouse tail-vein lung models of metastasis
All animal procedures were approved by the Brigham and Women's Hospital Institutional Animal Care and Use Committee. To understand whether B7-H3 plays a role in pulmonary LAM, Tsc2[-/-] 105K renal tumor cells (4.0 × 10[6]) or Tsc2[-/-] TTJ renal tumor cells (1.0 × 10[6]) stably expressing sh-NC (non-targeting control), sh-B7-H3 (1) or sh-B7-H3 (2) were injected into the lateral tail vein of 6-8-week-old male WT mice. Animal weight was monitored following injection of cells during the whole experimental course. Mice were harvested 4 weeks post-injection. Lungs were inflated at 25-cm H$_2$O pressure in 4% paraformaldehyde or with 1:1 optimal cutting temperature (OCT) in PBS and put into OCT, flash-frozen on dry ice. Tissues sections were cut at 5 μm thickness for H&E and immunohistochemistry. Lung tumor burden was quantified by the % of tumor occupied area in 5 lung sections at least 50 μm apart per mouse using Fiji ImageJ.

### Antibody-mediated depletion in vivo experiments
Mice were treated with 200 μg monoclonal anti-CD4 (GK1.5, BioXCell Cat#BE0003), anti-CD8 (YTS 169.4, BioXCell Cat#BE0117), isotype control rat IgG2b (LTF-2, BioXCell Cat#BE0090), or isotype control rat IgG1 (HRPN, BioXCell Cat#BE0088) antibodies by intraperitoneal injection 4 days prior tumor challenge and every 3 days later for a total of 13 treatments.

### Mass Cytometry (CyTOF) analysis
Mouse tumor samples were processed as previously described to obtain single-cell suspensions (Liu et al., 2018). Tumor-infiltrating immune cells were enriched in a Ficoll gradient as described previously (Liu et al., 2018). Cells were washed with CyTOF PBS and stained with cisplatin solution (1:2000, Fluidigm Cat#201064) for 2 min. Then, cells were washed twice with CyTOF PBS and stained with mouse FcR-Blocking reagent (1:100, Miltenyi Biotec Cat#130-092-575) at RT for 10 min followed by staining with metal-coupled extracellular antibodies cocktail (Supplementary Table 1) at RT for 30 min. Cells were washed twice and resuspended in 2% formaldehyde diluted from 16% formaldehyde (Electron Microscopy Cat#15710) and kept at 4 °C

overnight until acquisition day. Prior to the acquisition, cells were stained with Intercalator (1:2000, Fluidigm Cat#201103B) in CyTOF PBS for 10 min and washed once with CyTOF PBS, once with milliQH$_2$O and resuspended in Maxpar Cell Acquisition Solution (Fluidigm, Cat#201237) and analyzed using the CyTOF 3 Helios system (Fluidigm, instrument located at the Dana Farber Cancer Institute Mass Cytometry Core). Data were processed using Cytobank.

## Flow Cytometry

Single-cell suspensions were stained with antibodies against surface molecules (Supplementary Table 2) at RT for 30 min. For intracellular staining, cells were stimulated with cell stimulation cocktail (plus protein transport inhibitors) containing PMA, ionomycin, brefeldin A and monensin (1:500 dilution, eBioscience Cat#00-4975-93) at 37°C in RPMI media containing 10% heat-inactivated FBS (R&D systems, Cat#S11150H) and 0.05 mM 2-mercaptoethanol (ThermoFisher Scientific, Cat#21985023) for 4 hr prior to staining with Zombie NIR (BioLegend, Cat#423105). Next, cells were stained with antibodies against surface proteins followed by fixation and permeabilization and staining with antibodies against intracellular proteins. Samples were analyzed by a BD LSRFortessa SORP equipped to detect 17 or 15 fluorescent parameters. Compensation and data analysis were carried out using FACSDiva and FlowJo software. Gating strategy is shown in Supplementary Figs. 8 and 9.

## Ex vivo tumor cells isolation

$Tsc2^{-/-}$ 105K tumors stably expressing sh-NC (Non-targeting Control), sh-B7-H3 (1) or sh-B7-H3 (2) generated as described above in the in vivo experiments in mouse tumor models section were isolated and processed using the mouse tumor dissociation kit (Miltenyi Biotec, Cat#130-096-730) to achieve single-cell suspensions according to the manufacturer's protocol. Then tumor cells were isolated using a mouse tumor cell isolation kit (Miltenyi Biotec, Cat#130-110-187) according to the manufacturer's instructions. Briefly, single-cell suspensions were incubated with 20 µL of non-tumor cell depletion cocktail per $10^7$ cells in total for 15 min. The unlabeled cells that passed through LS columns (Miltenyi Biotec, Cat#130-042-201) placed in the magnetic field of the MACS Separator (Miltenyi Biotec) were collected.

## TCGA cancer patient dataset analysis

**File generation.** Manifests containing fragments per kilobase per million (FPKM) normalized RNA-seq data from 34 TCGA cohorts – Acute Myeloid Leukemia - (TCGA-LAML), Adrenocortical carcinoma (TCGA-ACC), Bladder Urothelial Carcinoma (TCGA-BLCA), Glioblastoma multiforme and Brain Lower Grade Glioma and (TCGA-GBMLGG), Breast Invasive Carcinomas (TCGA-BRCA), Cervical Squamous Cell Carcinoma and Endocervical Adenocarcinoma (TCGA-CESC), Cholangiocarcinoma (TCGA-CHOL), Chronic Myelogenous Leukemia (TCGA-LCML), Colon Adenocarcinoma (TCGA-COAD), Esophageal Carcinoma (TCGA-ESCA), Head and Neck Squamous Cell Carcinoma (TCGA-HNSC), pan-Kidney Cancer (TCGA-KIPAN), Liver Hepatocellular Carcinoma (TCGA-LIHC), Lung Adenocarcinoma (TCGA-LUAD), Lung Squamous Cell Carcinoma (TCGA-LUSC), Lymphoid Neoplasm Diffuse Large B-cell Lymphoma (TCGA-DLBC), Mesothelioma (TCGA-MESO), Ovarian Serous Cystadenocarcinoma (TCGA-OV), Pancreatic Adenocarcinoma (TCGA-PAAD), Pheochromocytoma and Paraganglioma (TCGA-PCPG), Prostate Adenocarcinoma (TCGA-PRAD), Rectum Adenocarcinoma (TCGA-READ), Sarcoma (TCGA-SARC), Skin Cutaneous Melanoma (TCGA-SKCM), Stomach Adenocarcinoma (TCGA-STAD), Testicular Germ Cell Tumors (TCGA-TGCT), Thymoma (TCGA-TGCT), Thyroid Carcinoma (TCGA-THCA), Uterine Carcinosarcoma (TCGA-USC), Uterine Corpus Endometrial Carcinoma (TCGA-UCES), Uveal Melanoma (TCGA-UVM) were downloaded from the Broad Institute GDAC (TCGA data version 20150601).

Patients were then sorted by increasing B7-H3 expression in each cohort.

**Generation of mTORC1 score.** The 199 genes from Hallmark mTORC1 signaling gene set in the molecular signature database[85] was used to generate an "mTORC1 score" for each patient in the 34 TGCA cohorts by conducting single sample gene-set enrichment analysis (ssGSEA) using Gene Set Variation Analysis (GSVA)[86].

**mTORC1 score and *CD276* expression.** Each cancer type was analyzed separately to set high-low cut point for mTORC1 score and *CD276* expression. Patients within the 20th percentile of *CD276* expression and/or mTORC1 score were designated as the "low expression" subgroup, while patients above the 80% percentile of *CD276* expression and/or mTORC1 score were designated as the "high expression" subgroup. All patients within the low *CD276* expression group were further stratified for 50% with low mTORC1 score and were designated the "low *CD276* and low mTORC1 score" subgroup. All patients within the high *CD276* expression group were further selected for 50% with high mTORC1 score and were designated the "high *CD276* and high mTORC1 score" subgroup. The log-rank test was used to compare the survival difference of two groups at each observed time. The Kaplan-Meier analysis was applied to obtain a survival-curve plot for all cancer types, excluding chronic myelogenous leukemia (LAML) due to the unavailability of the survival data. The hazard ratio was assessed using the COX proportional hazards model with Efron's approximation.

**Six immune subtypes analysis.** For analysis of mTORC1 score and *CD276* expression with the immune microenvironment, six immune subtypes metrics were downloaded from Thorsson et al. 2018[40]. We analyzed all 34 cancer types as described above.

## RNA-seq

Total RNA isolated from ex vivo tumor cells or bulk tumors was extracted using the RNeasy Plus Micro Kit (Qiagen Cat#74034) or RNeasy Mini kit (Qiagen Cat#74106) according to the manufacturer's instructions. RNA from each group was sequenced at the Molecular Biology Core Facilities at Dana-Farber Cancer Institute. The quantity and quality of mRNA was examined using an Agilent 2200 TapeStation instrument and by SYBR qRT-PCR assay. mRNA was captured using magnetic oligo-dT beads, and subsequently used to make cDNA libraries using the KAPA mRNA HyperPrep Kit (Roche), then sequenced using single-end 75cycle (HO) on an Illumina NextSeq500 instrument (Illumina). Differential gene expression analysis on three biological samples was performed using the VIPER pipeline by the Molecular Biology Core Facilities at Dana-Farber Cancer Institute. For GSEA, the gene expression was processed and analyzed by GenePattern using ExpressionFileCreator Module (version 11.14) and GSEA module (v14).

## Single-cell dissociation and CITE-seq

Tumors were dissociated using a modified version of a published protocol described in ref. [87]. To sort intratumoral live CD3$^+$ T cells, 80 sh-NC and 80 sh-B7-H3 (1) tumors (size ~300-400mm$^3$) were used. Minced tumor fragments from every 300-400 mm$^2$ of tumor were added to 3 mL of protease solution [5 mM CaCl2, 10 mg/ml Bacillus Licheniformis protease (Sigma, Cat#P5380) and 125 U/mL DNase I recombinant (Roche, Cat#4716728001) in PBS and incubated with gentle rotation at 4 °C for 10 min. After incubation, the solution was transferred to a Miltenyi C-tube (on ice) and the Miltenyi gentleMACS brain_03 program was run three times in a cold room (4 °C). The incubation was repeated with gentle rotation at 4 °C for 10 min. Single-cell dissociation was confirmed by microscopic examination. 3 mL 0.25% Trypsin-EDTA was added to the cell suspension and incubated at RT for 1 min. This was followed by the addition of 3 mL ice-cold PBS with 10% FBS. Cells were passed through a 70 µM strainer and the filter

rinsed with 3 mL ice-cold PBS with 10% FBS. Tumor-infiltrating immune cells were separated from other cells by centrifugation with the deceleration brake set at 1 at 1260 g at 4 °C for 25 min in a Ficoll gradient (GE Healthcare, Cat#17-1440-03). Cells were stained with Mouse TrueStain Plus FcX (Biolegend, Cat#156604) at 4 °C for 10 min followed by staining with an anti-CD3 antibody (Biolegend, Cat#100204) for 30 min at 4 °C. Tumors from each group were pooled together, then $1 \times 10^5$ live CD3$^+$ T were sorted by flow cytometry and stained with Mouse TotalSeq-A CITE-seq antibodies (Biolegend, # 99833) at 4 °C for 30 min, before being washed in excess PBS with 2% FBS and 2.5 mM EDTA and resuspended in PBS with 0.04% BSA at 1500 cells/ml.

CITE-seq libraries and sequencing were done by the BWH Center for Cellular Profiling. Single-cell suspensions were super-loaded onto a single lane (Chromium chip G, 10X Genomics). After generating single-cell Gel Bead-in-Emulsions (GEMs), cDNA from poly-adenylated mRNA and DNA barcodes from cell surface CITE-seq antibodies were generated simultaneously from the same single cell inside the GEM. Gene expression libraries were prepared using Chromium Next GEM Single-Cell 3' Gel Bead and Library Kit (10x Genomics, V.3.1) according to the manufacturer's instructions. Briefly, GEM-RT was performed in a 96-Deep Well Reaction Module: 53 °C for 45 min, 85 °C for 5 min; end at 4 °C. After RT, GEMs were broken down, then the cDNA and DNA barcodes were cleaned up with DynaBeads MyOne Silane Beads. cDNA was amplified with 96-Deep Well Reaction Module: 98 °C for 15 s, 67 °C for 20 s, and 72 °C for 1 min; 72 °C for 1 min; end at 4 °C. Followed by size selection to separate the amplified cDNA molecules for 3' Gene Expression and Cell Surface Protein library construction SPRIselect Reagent. Indexed sequencing libraries were generated using the reagents in the Single-Cell 3' Library Kit. Gene expression and ADT (antibody-derived tag) libraries were analyzed using the Agilent 2100 Bioanalyzer. Gene expression libraries were sequenced on an average of 50,000 reads per cell and ADT libraries were sequenced on an average of 10,000 reads per cell on a Novaseq 6000 system (Illumina).

### Preprocessing and analysis of CITE-seq Data
FastQ files from the 10x libraries were processed with Cell Ranger v6.0.1., which supports analysis of cell multiplexing data for the 3' Gene expression. Quantification was performed using the STAR aligner against the mmu10 transcriptome. Normalized and downstream analyses of protein and RNA data were performed using the Seurat R package (version 4.0)[88], which enables the integrated processing of multi-modal single cell datasets. The protein tag and RNA raw counts were processed separately first by the Seurat's reciprocal PCA (RPCA) fast integration pipeline. Protein tag raw counts were normalized using centered log ratio (CLR) transformation, where counts were divided by the geometric mean of the corresponding tag across cells and log-transformed[89]. For the RNA count matrices, after we filtered out low-quality cells (<200 and >3000 unique genes, >5% mitochondrial reads) and doublets, 12,820 CD3$^+$ tumor-infiltrating lymphocytes were further analyzed. Then, we obtained the highly variable genes and proteins, corrected, logarithmically normalized, and scaled the counts (default parameters) of each data set using Seurat (4.0). To define the significant difference of target gene expression changes in cell clusters, we used *FindAllMarkers* with maximum cell number ranging from 10 to 100, in 10 sampling iterations for each cell number. For the clustering, the first 40 principal components were used and 0.8 was used as the resolution parameter. The top 20 genes upregulated in each cluster (*FindAllMarkers*) were used to label the cluster.

### Differential gene expression analysis and GSEA from CITE-seq data
We performed differential expression analysis to identify the genes in tumor-infiltrating CD4$^+$ and CD8$^+$ T cells associated with B7-H3

knockdown tumors. We used DESeq2 package for analysis of aggregated read counts[90]. We first aggregated reads across biological replicates of tumor infiltrating CD4$^+$ and CD8$^+$ T cells from control and B7-H3 knockdown tumors, transforming a genes-by-cells matrix to a genes-by-replicates matrix using matrix multiplication. Then we used a Wald test of negative binomial model coefficient and likelihood ratio test compared to a reduced model to compute the statistical significance. KEGG and GSEA were performed with clusterProfiler package[91,92], which supports statistical analysis and visualization of functional profiles for gene clusters.

### Tumor infiltrating T-cell isolation and culturing
Single-cell suspensions from mouse tumors were processed as previously described to obtain single-cell suspension[43]. Single-cell suspensions were stained and isolated by FACS. CD4$^+$ TILs (CD45$^+$CD3$^+$CD4$^+$CD25$^-$CD38$^+$CD39$^+$) were sorted into ATCC modified RPMI medium in the presence of 10% FBS. CD45$^+$CD3$^+$CD4$^+$CD25$^-$CD38$^+$CD39 TILs were pooled together, centrifuged and resuspended in ATCC modified RPMI medium (10% FBS) with addition of Dynabeads Mouse T-Activator CD3/CD28 (GIBCO, Cat#11452D) for culturing. T cells were cultured in 96 well U-bottom plates, and briefly centrifuged to ensure good cell contact with Dynabeads. T cell activation was conducted in two phases. For the first week, T cells were cultured in ATCC modified RPMI medium (10% FBS) plus 200 IU/ml mouse recombinant IL-2 (BioLegend, Cat#575402). Starting from the second week, the IL-2 concentration was increased to 6000 IU/ml and T cells were harvested between 2 weeks for killing assays.

### Cytotoxic T cell killing assay
To achieve an effector-to-target (E:T) ratio of 10:1, 10,000 Tsc2-deficient 105K cells were plated in 96 well plates, pre-treated with 10 ng/ml of mouse IFN-γ (Miltenyi Biotech, Cat#130-105-774), and Incucyte Nuclight Rapid Red Dye (1:500 dilution, Sartorius, Cat#4717) for 48 hr before addition of tumor-infiltrating T cells. Each well contained 100 ul of medium supplemented with 1 μl/well of Incucyte Cytotox Green Dye (Sartorius, Cat#4633). For MHC-II blocking experiments, 10 μg/ml of isotype control antibody (BioLegend, Cat#400601) or MHC-II antibody (BioLegend, Cat#107601) was added to each well pre-plated with Tsc2-deficient 105K cells and cultured for 2 hr before addition of CD45$^+$CD3$^+$CD4$^+$CD25$^-$CD38$^+$CD39$^+$ TILs. Cell culture was monitored by the IncuCyte S3 Live Imaging system (Sartorius) at 1-hr intervals for up to 48 hr when needed. All experiments were carried out with samples from each group pooling from 10 independent tumors. The number of dying tumor cells was determined by co-localization of the Cytotox Green signal to tumor cells. Any out of focus frames were discarded.

### Direct and Indirect co-culture of T cells with Tsc2-deficient tumor cells
To study the impact of B7-H3 in Tsc2-deficient cells on T cells, we performed co-cultures. For direct co-culture experiments, sh-NC, sh-B7-H3 (1), sh-B7-H3 (2), sgCtrl, sg*Cd276* (1), or sg*Cd276* (2) Tsc2-deficient 105K cells were plated in 6-well plates in DMEM supplemented with 10% FBS. Splenic CD4$^+$ or CD8$^+$ T cells were isolated from naïve WT C57BL/6 J mice using MojoSort Mouse CD4 T cell isolation Kit or MojoSort Mouse CD8 T cell isolation Kit, respectively and activated using Dynabeads Mouse T-Activator CD3/CD28 (GIBCO, Cat#11452D) in RPMI medium plus 10% FBS for 16 hr before adding to tumor cells at a tumor-to-T-cell (E:T) ratio of 1:2 for 48 hr. For indirect co-culture experiments, conditioned media from sh-NC, sh-B7-H3 (1), or sh-B7-H3 (2) Tsc2-deficient 105K cells were harvested after 24 hr of serum-starvation. Conditioned media were concentrated using Amicon Ultra-4 Centrifugal Filter Unit (Millipore Cat#UFC800324) and 200 μg of protein was added to isolated activated splenic CD4$^+$ or CD8$^+$ T cells ($2\times10^5$) as described above for 48 hr before analysis by

FACS. Cells were processed and stained as described in the Flow Cytometry section.

## Immunohistochemistry

FFPE human tissue microarray slides were purchased from US Biomax, Cat#MC801, and Novus Biologicals, Cat#NBP2-42052 with ethical approval by the Partners Human Research Committee of Brigham and Women's Hospital. Written informed consent was obtained from participants. Tissues were dewaxed in 3 changes of xylene followed by 3 changes of 100% ethanol. Heat-induced antigen retrieval was performed in a Russell Hobbs pressure cooker for 10 min in sodium citrate antigen retrieval solution (10 mM sodium citrate, pH 6.0). Sections were blocked in 5% normal goat serum for 60 min, followed by incubation with B7-H3 (1:200, CST, Cat#14058), CD4 (1:1000, eBioscience, Cat#14-9766-82), CD8 (1:1000, eBioscience, Cat#14-0808-82), pS6 S235/236 (1:400, CST, Cat#2211), pS6 S240/244 (1:250, CST, Cat# 5364), p-p70 S6 Kinase T389 (1:50, CST, Cat#9205), E-cadherin (1:400, CST, Cat#114472), STAT1 (1:100, CST, Cat#14994), CD31 (1:200, Abcam Cat#ab182981), or F4/80 (1:400, CST Cat#70076) antibodies at 4° C overnight. Sections were incubated with polymer anti-mouse IgG (Vector Laboratories, Cat#MP7452) or polymer anti-rabbit IgG (Vector Laboratories, Cat#MP-7451) at RT for 30 min. Signal was detected using DAB substrate, Peroxidase (Vector Laboratories, Cat#SK-4105) according to manufacture's instructions. Slides were counterstained with hematoxylin (Agilent, Cat#S330930-2) and mounted with D.P.X. For immunofluorescent co-staining, FFPE sections were stained with the indicated primary antibodies as above then with anti-rabbit or anti-mouse IgG secondary antibody, Alexa Flour 488 or 594 (1:500) for 45 min at RT. Representative images were taken using a Keyence fluorescence microscope or Olympus FV10-FWS confocal microscope and quantified by counting the positive cells of five different fields/tumor using Fiji ImageJ.

Frozen tissues sections were fixed with cold methanol for 15 min. Sections were blocked and stained with E-cadherin (1:400, CST Cat#114472), MHC-II (1:100, LSBio Cat#LS-C204829), CD4 (1:100, Biolegend, Cat#100424), CD38 (1:100, Biolegend, Cat#102725), or CD39 (1:250, Abcam, Cat#Ab227840). Sections were incubated with anti-rabbit or anti-mouse IgG secondary antibody, Alexa Flour 488 or 594 (1:500) for 45 min at RT, countered stained with 1 μg/mL DAPI (Sigma, Cat#D9542), mounted with Fluoromount G (SouthernBiotech, Cat#0100-01), and imaged and quantified as describe above.

## Tissue microarray pathology-based scoring

The manual scoring procedure was performed with digital scans of the tissue microarrays using images taken by the Keyence digital microscope (BZ-X800). The immunoreactivity score was determined by two independent observers, who assessed the relative amounts of stained cells and the staining intensity. The observers were blinded during the scoring procedures. For assessment of tumor cells, an ordinal scale was used based on the number of stained cells and staining intensity (amount of stained cells: 0 = no stained cells, 0%; 1 =< 10%; 2 = 10–50%; 3 = 51–80%; 4 => 80% staining, intensity: 0 = no staining; 1 = mild brown; 2 = moderate brown; 3 = intense brown). The total score was obtained by multiplying the staining intensity and percentage scores.

## siRNA transfections

The following siRNAs (GE Dharmacon SMARTpools ON-TARGET plus) were transfected at 12.5 nM using Lipofectamine RNAiMAX (Thermo-Fisher Scientific, Cat#13778150) according to manufacturer's instructions: mTOR (L-065427-00-0005), Raptor (L-058754-01-0005), Rictor (L-064598-01-0005), S6K (L-040893-00-0005), 4E-BP1 (L-058681-01-0005), YY2 (L-171481-00-0005) or control (Non-targeting siRNA pool D-001810-10-05).

## Plasmid DNA transfections, virus production and transduction

Plasmids encoding wild-type YY2 (Myc-DDK-YY2, Cat#MR218884) was purchased from Origene, USA, wild-type S6K (pRK7-HA-S6K1-WT, Cat# 8984), kinase-dead S6K (pRK7-HA-S6K1-KR, Cat# 8985), wild-type Smurf1 (pCMV5B-Flag-Smurf1 wt, Cat# 11752), wild-type mTOR (pcDNA3-FLAG-MTOR, Cat#26603), L1460P mutant mTOR (pcDNA3-FLAG-MTOR-L1460P, Cat#69006), C1483F mutant mTOR (pcDNA3-FLAG-MTOR-C1483F, Cat#69008), S2215Y mutant mTOR (pcDNA3-FLAG-MTOR-S2215Y, Cat#69013), and R2505P mutant mTOR (pcDNA3-FLAG-MTOR-R2505P, Cat#69015) were purchased from Addgene, USA. Lentiviral plasmids encoding human wild-type eGFP-YY2 and human mutant T336A eGPF-YY2 were custom made by VectorBuilder. For immunoblot or immunofluorescent analysis, cells were transfected using Lipofectamine 3000 as recommended by the manufacturer (Invitrogen) in full-growth media for 24 hr prior lysis or fixation.

Lentiviral particles containing plasmids of wild-type eGFP-YY2 or T336A eGFP-YY2 were produced by transfecting HEK293T cells with 3rd generation lentiviral packaging plasmids, including pMDLg/pRRE (7.5 μg), PRSV/REV (7.5 μg), pMD2.G (5 μg) and the lentiviral plasmids of wild-type eGFP-YY2 or T336A eGFP-YY2 (15 μg) for 16 hr with Lipofectamine 3000 as recommended by the manufacturer (Invitrogen). Cells were further incubated in fresh full-growth media for another 48 hr and the media containing lentivirus were collected after incubation.

HeLa cells stably expressing wild-type eGFP-YY2 or mutant T336A eGFP-YY2 were generated by transducing wild-type eGFP-YY2 or mutant T336A eGFP-YY2 lentivirus packaged as described above. Cells were selected in complete growth medium containing 0.5 μg/ml puromycin for 1 week and sorted for GFP-positive cells using FACS then maintained in 0.25 μg/ml puromycin.

## Crystal violet proliferation and anchorage-independent cell growth assays

Cells were seeded ($1 \times 10^3$ cells per well) in 96-well plates in growth media on day 0. At the time of harvest cells were fixed in 10% formalin, stained with 0.5% crystal violet (Sigma, Cat#V5265). Crystal violet-stained cells were dissolved in 100% methanol as described previously[43]. For anchorage-independent cell growth assays, $0.5 \times 10^4$ cells were plated in six-well 35 mm tissue culture dishes containing a bottom layer of 0.7% (w/v) agarose in complete medium and a top layer of 0.3% agarose solution mixed with cells and incubated at 37°C for 3 weeks. Colonies were imaged using an Olympus SZH10 Stereo microscope (Olympus, Japan) and a Nikon D3000 camera (Nikon, Japan).

## Transduction of shRNAs

Heterogenous pools of B7-H3 knockdown 105K or TTJ cells were generated by transducing non-target shRNA (Sigma, Cat#SHC016V) or two shRNA sequences targeting two distinct regions of B7-H3 (CD276) mRNA (Sigma, Cat#TRCN0000246370, TRCN0000246371). Clones were selected in complete growth medium containing 1 μg/ml puromycin for 1 week and then maintained in 0.5 μg/ml puromycin.

## Generation of B7-H3/CIITA/IFNGR1 KO cells using CRISPR/Cas9

A DNA construct consisting of a pool of three guide-RNAs targeting Cd276 in the CRISPR/Cas9 knockout expression vector (Santa Cruz, Cat#sc-430440) or control CRISPR/Cas9 plasmid (Santa Cruz, Cat#sc-418922) was transfected into Tsc2-deficient 105 K cells. 48 hr after transfection, the transfected cell population was sorted into single cells which were analyzed for GFP expression by flow cytometry and expanded in culture. For CIITA and IFNGR1, a DNA construct consisting of a pool of two-guide RNAs targeting Ciita or Ifngr1 along with scramble were custom-made by VectorBuilder. The small guide

sequences used for scramble sgRNAs are 1. GTGTAGTTCGAC-CATTCGTG, 2. GTTCAGGATCACGTTACCGC; *Ciita* sgRNAs are 1. TCATTGCTTGGATCGTCCCG, 2. TCCCGGAGCCTTAGTCGAGC; *Ifngr1* sgRNAs are 1. ATGTGGAGCATAACCGGAGT, 2. TGGTATTCCCAGCA-TACGAC. The loss of B7-H3/CIITA/IFNGR1 protein in the clones was confirmed via western blotting analysis.

## Immunoblotting

Cells were washed with ice-cold PBS then lysed in RIPA lysis buffer containing protease inhibitor cocktail (Sigma, Cat#P8340) and phosphatase inhibitor cocktail (Sigma, Cat# P0044). Total cell lysates were rotated at 4 °C for 30 min and centrifuged at 4 °C for 10 min. Total protein was separated via 4–12% SDS-PAGE and transferred to PVDF membranes. The membranes were blocked with TBS containing 0.1% Tween-20 and 5% bovine serum albumin and then incubated with primary antibodies overnight at 4 °C. After incubation with the appropriate horseradish peroxidase-conjugated secondary antibodies, the membranes were developed using SuperSignal™ West Pico PLUS Chemiluminescent Substrate (ThermoFisher Scientific, Cat#34580) and the images were captured using iBright Western Blot Imaging systems (ThermoFisher Scientific). The following primary antibodies were used for immunoblotting at 1:1000 dilution unless otherwise indicated: Phospho-p70 S6 Kinase T389 (CST, Cat#9234), p70 S6 Kinase (CST, Cat#2708), TSC2 (CST, Cat#4308), Phospho-S6 S235/236 (CST, Cat# 2211), S6 (CST, Cat#2217), mTOR (CST, Cat# 2983), Raptor (CST, Cat#2280), Rictor (CST, Cat#2114), Phospho-Akt S473 (CST, Cat#4060), Phospho-Akt T308 (CST, Cat#4056), Akt (CST, Cat# 4685), GAPDH (CST, Cat# 2118), CREB (CST, Cat#4820), STAT1 (CST, Cat#14994), p-STAT1 (Tyr 701) (CST, Cat#9167), human B7-H3 (CST, Cat# 14058), YY1 (CST, Cat#46395), HA-tag (CST, Cat#3724), Phospho-substrate (RXXS*/T*) (CST, Cat# 9614), MHC-II (1:500, LSBio Cat#LS-C204829), β-actin (Sigma, Cat#A1978), YY2 (A-5) (1:500, Santa Cruz, Cat#sc-377008), YY2 (C-10) (1:500, Santa Cruz, Cat#sc-374455), mouse B7-H3 (1:500, R&D Systems, Cat#AF1397), and GFP (Abcam, Cat#ab6556).

## Co-immunoprecipitation

HeLa cells stably expressing wild-type eGFP-YY2 were transfected with Smurf1-FLAG for 24 hr using Lipofectamine 3000 as recommended by the manufacturer. Cells were harvested, washed with PBS and lysed in home-made Co-IP buffer (25 mM Tris-HCl, pH 7.4, 150 mM NaCl, 1% NP-40, 1 mM EDTA, 5% glycerol) containing protease and phosphatase inhibitor cocktails at 4 °C for 50 min with rotation and centrifuged at 14,000 × g at 4 °C for 10 min. Cell lysates were rocked with 50 µl of anti-GFP antibody-magnetic beads (MBL, Cat#D153-11) at 4 °C for 4 hr. Th beads were washed four times in home-made Co-IP buffer. Bound proteins were boiled and analyzed by SDS/PAGE followed by immunoblot analysis.

## In vitro Kinase assay

Recombinant p70 S6 Kinase protein (2 µg, Millipore, Cat#14-486) was incubated with DYKDDDK-YY2 (2 µg, Euprotein, Cat#EP8149080) or DYKDDDK-eIF4B (2 µg, positive control, Euprotein, Cat#EP8052870) in 1 x kinase buffer (CST, Cat#9802) with 400 µM of ATP at 30 °C for 1 hr. The reactions were gently tapped every 10 min, and the reactions were stopped by adding 2x Sample buffer and then resolved by SDS-PAGE. Phosphorylation of DYKDDDK-YY2 and eIF4B was detected by Western blotting using the Phospho-Substrate (RXXS*/T*) (110B7E) (CST, Cat#9614) or p-eIF4B S422 (CST, Cat#3591), respectively.

## Phosphorylation assay

HeLa cells stably expressing wild-type eGFP-YY2 or mutant T336A eGFP-YY2 were treated with 40 µM PF-4708971 or vehicle for 1 hr before cell lysis with 1 x RIPA buffer (CST, Cat# 9806 S). Cell lysates were harvested and rocked with 50 µl of anti-GFP antibody-magnetic beads (MBL, Cat#D153-11) at 4 °C for 4 hr. The beads were washed four times. Bound proteins were boiled and analyzed by SDS/PAGE followed by immunoblot analysis using the ani-phospho-Substrate (RXXS*/T*) antibody (CST, Cat#9614).

## Ubiquitination assay

HeLa cells stably expressing wild-type eGFP-YY2 or mutant T336A eGFP-YY2 were treated with 30 µM PF-4708971 or transfected with plasmids expressing WT-HA-S6K and Smurf1-FLAG for 24 hr, and then treated with MG132 at 20 µM for 5 hr before cell lysis with 1 x RIPA buffer (CST, Cat#9806S). Cell lysates were harvested and rocked with 50 µl of anti-GFP antibody-magnetic beads (MBL, Cat#D153-11) at 4 °C for 4 hr. The beads were washed four times. Bound proteins were boiled and analyzed by SDS/PAGE followed by immunoblot analysis.

## Immunocytochemistry and Immunofluorescence

Cells were fixed in 4% PFA for 10 min, rinsed in PBS, then permeabilized with 0.5% Triton X-100 in PBS for 20 min. Then, cells were blocked with 1% BSA for 30 min, and incubated with YY2 (Santa Cruz, Cat#sc-377008) and/or pS6 (CST, Cat# 2211) for 2 hr. The primary antibodies were detected by anti-rabbit IgG Secondary Antibody Alexa Flour 594 for 45 min, counter stained with 1 µg/mL DAPI (Sigma, Cat#D9542), mounted with Fluoromount G, and imaged as described above.

## Cytoplasmic and Nuclear Protein Fractionation

Cells were harvested and washed twice in cold PBS. The cytoplasmic and nuclear protein were isolated using the Nuclear/Cytosol Fractionation kit (BioVision, Cat#K266-25) according to the manufacturer's instructions. Equal amounts of proteins were run on SDS-PAGE gels and immunoblotted.

## Chromatin-immunoprecipitation (ChIP) Followed by qPCR (ChIP-qPCR)

ChIP-qPCR was performed by using the High-Sensitivity ChIP Kit (Abcam, #ab185913) according to the manufacturer's protocol. In brief, *Tsc2* KO MEFs or 105K cells were crosslinked by 1% formaldehyde and chromatin was extracted and sheared by a M220 Focused-ultrasonicator (Covaris). Samples were immunoprecipitated with anti-YY2 antibody (Santa Cruz Cat#sc-377008). The primer sequences used for ChIP-qPCR are listed in the Supplementary Table 3. The results were from 3 biological samples followed by normalization to input signals and showed as mean ± SD.

## Dual luciferase Assays

*Tsc2*-WT MEFs, *Tsc2* KO MEFs, 105K or 105K + TSC2 cells were seeded in 96 well plates and transfected with 250 ng dual luciferase plasmid psiCHECK2 (Promega, Cat#C8021) or psiCHECK2 plasmid containing *Cd276* promoter for 24 hr. For siRNA transfection luciferase assays, cells were transfected with 6 pmol of the siRNA listed in Supplementary Table 4 using Lipofectamine RNAiMAX. Twenty-four hours after transfection with siRNA, cells were transfected with psiCHECK2 or psiCHECK2 plasmids containing *Cd276* promoter and incubated for additional 48 hr. Cell lysates were analyzed using the Dual-Glo Luciferase Reporter Assay (Promega, Cat#E2920). Firefly luciferase activity was normalized to Renilla luciferase activity.

## RNA isolation and RT-PCR

Total RNA from cultured cells or tumor cells was extracted using the RNeasy Mini kit (Qiagen, Cat#74106) or RNeasy Plus RNA Micro Kit (Qiagen, Cat#74034). Extracted total RNA was reverse transcribed into cDNA using High-Capacity cDNA Reverse Transcription kit (Thermo-Fisher Scientific, Cat#4368814). Quantitative real-time PCR (qPCR) was performed using Taqman Fast Advanced Master Mix (ThermoFisher Scientific, Cat#4444963). Real-time PCR was performed using an

Applied Biosystems StepONE Plus Real-time PCR system (Thermo-Fisher Scientific) with gene-specific primers (Supplementary Table 5). Values represent the average of three independentexperiments normalized to β-actin.

## Proximity Ligation Assay (PLA)

PLA assays were performed using the Duolink in situ red starter kit for proximity ligation assays (Sigma Aldrich, Cat#DUO92101) following the manufacturer's instructions. Briefly, cells grown on chamber slides were fixed in 4% PFA for 10 min and permeabilized in ice-cold methanol for 20 min, then blocked with blocking buffer supplied with the kit at 37 °C for 1 hr. Cells were incubated with primary antibody (mouse anti-YY2, Santa Cruz, Cat#374455, 1:100, and rabbit anti-p70 S6K, CST, Cat#2708, 1:100) overnight at 4 °C, and then with the provided secondary antibody (conjugated with nucleotides) for 1 hr at 37 °C with washes after each step. Ligation of the nucleotides and amplification of the strand occurred sequentially by incubating cells first with ligase and then with polymerase and detection solution. Mouse anti-YY2 with normal rabbit IgG, rabbit anti- anti-p70 S6K with normal mouse IgG or normal rabbit and mouse IgG were used as negative controls. Representative images were taken using a Keyence fluorescence microscope and quantified by counting the red fluorescent signals in the nucleus of cells using Fiji ImageJ.

## Quantification and Statistical Analysis

All experiments were performed at least three times. Results are expressed as mean ± SD/SEM, and analyzed by two-tailed unpaired Student's $t$-test, Mann-Whitney $U$ test, nonparametric Kruskal-Wallis test followed by Dunn's multiple comparisons test, one-way ANOVA followed by Dunnett's/Tukey's comparisons test or two-way ANOVA followed by Holm-Sidak's multiple comparisons test. For Kaplan-Meier survival analyses, log-rank testing was used to evaluate the difference between curves. Hazard ratio was determined by COX proportional hazards regression with Efron's approximation. Correlation of protein expression analyzed by immunohistochemistry was assessed by Pearson correlation coefficient. The $p$ value <0.05 was considered statistically significant. The analysis was conducted using Graphpad 6.0 and 9.3.0 software.

## Reporting summary

Further information on research design is available in the Nature Portfolio Reporting Summary linked to this article.

## Data availability

RNA-seq and CITE-seq data generated in this study have been deposited in the GEO database under accession code GSE213626 and GSE213939, respectively. The TCGA publicly available data are available in the FireBrowse. All other data supporting the finding of this study are available within the article and the data generated in this study are provided in the Supplementary Information. Source data are provided with this paper.

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

## Acknowledgements

We thank Zhu Zhu and Adam Chicoine from the BWH Center for Cellular Profiling for technical advice. We thank Drs. Hans Widlund, Brendan Manning, Wenyi Wei, and XiaoMing Dai for valuable discussions. We thank Dr. Michael Hall for providing the inducible Raptor and Rictor knockout MEFs. We thank Dr. Vera Krymskaya for providing TTJ cells. H.J.L. is a recipient of the U.S. Department of Defense Exploration Hypothesis Development Award (W81XWH-21-1-0223) and The LAM Foundation Career Development Award (LAM0149C01-21). This study was supported by the Engles Family TSC/LAM Research Fund, the Lee Family Research Fund, and National Institutes of Health (NIH) U01 grant 1U01HL131022-01.

## Author contributions

H.J.L. Conceptualization; directed the project; developed experimental protocols; designed, performed, interpreted, and analyzed experiments; developed computational methods and performed bioinformatic analyses; wrote and edited manuscript. H.D., D.K. Contributed to performing experiments and edited manuscript. M.Z. Contributed to the transcription factor analyses. K.S.W. Contributed to CITE-seq experimental design. G.J.F. Discussion, resources, reviewed and edited manuscript. D.J.K. Discussion and edited manuscript. E.P.H. Supervised project, discussed and edited manuscript.

## Competing interests

G.J.F. has patents/pending royalties on the PD-1/PD-L1 pathway from Roche, Merck MSD, Bristol-Myers-Squibb, Merck KGA, Boehringer-Ingelheim, AstraZeneca, Dako, Leica, Mayo Clinic, and Novartis. G.J.F. has a patent on RGMb in cancer immunotherapy. G.J.F. has served on advisory boards for Roche, Bristol-Myers-Squibb, Xios, Origimed, Triursus, iTeos, NextPoint, IgM, Jubilant, Trillium, GV20, IOME, and Geode. G.J.F. has equity in Nextpoint, Triursus, Xios, iTeos, IgM, Trillium, Invaria, GV20, and Geode. The remaining authors declare no other competing interests.
