## [Peer Review File · Nature Communications]

mTORC1 upregulates B7-H3/CD276 to inhibit antitumor T cells
and drive tumor immune evasionEditorial Note: Figure s3m and Figure 7a in this Peer Review File were created by Biorender.com - ©BioRender.

REVIEWER COMMENTS

Reviewer #1 (Remarks to the Author): with expertise in B7-H3, cancer immunology

In the manuscript entitled “mTORC1 upregulates B7-H3/CD276 to inhibit antitumor T cells and drive tumor immune evasion”, the authors showed the correlation of both B7-H3 expression and mTORC1 activity with worse clinical outcomes by analysis of TCGA human tumors. In addition, they also found that B7-H3 expression was regulated by mTORC1 and blockade of B7-H3 pathway suppressed tumor growth with increased T cell activity and IFN γ responses. Of note, they also found highly cytotoxic CD38+CD39+CD4+ T cells in B7-H3 KO tumors. Although the manuscript shows the interaction between B7-H3 and mTORC1, it is not of high quality to warrant publication in this journal. Listed below are the major concerns with this manuscript.

Major concerns:

1. The paper “Tumor-expressed B7-H3 mediates the inhibition of antitumor T-cell functions in ovarian cancer insensitive to PD-1 blockade therapy” already showed that B7-H3 expressed on tumor cells, but not host cells, had a dominant role in suppressing antitumor immunity, which was dependent on CD8+ T cells.
2. The figures are not well prepared. For example, Figure 2f and Figure 2l are missing? Figure 1m should be figure 1l and Figure 1n is Figure 1m. In Figure 2n, what does each color represent? Extended Data Fig. 3c, d did not label cell type. Extended Data Fig. 7f cluster annotations of CD4-CP and CD4-GZMK were wrong in the color.
3. Most tumor cells are Tsc2-wild type, and B7-H3 protein is also expressed in Tsc2 wild-type MEFs (Fig. 2m). The authors should at least discuss the regulator of B7-H3 expression in TSC2-WT tumor.
4. B7-H3 expression is lower in sh-B7H3(2) group than that in sh-B7H3(1) group (Fig.6d), however, the tumor volume in sh-B7H3(2) group is bigger than that in sh-B7H3(1) group (Fig.5b) in WT mice and the percentage of CD4+ T cells is lower in sh-B7H3(2) group than that in sh-B7H3(1) group (Fig.5i). The authors should also analyze the correlation between B7-H3 knockdown efficiency with tumor growth.

Reviewer #2 (Remarks to the Author): with expertise in cancer immunology

Liu et al reported in this manuscript that mTORC1 critically controls tumor cell expression of B7-H3/CD276, which functions as a co-inhibitory molecule to suppress antitumor T cell response and evade immunosurveillance. They show that mTORC1 phosphorylates the transcriptional factor YY2 via S6K, preventing degradation of YY2 from Smurf1-mediated ubiquitination. YY2 binds to CD276 promoter and transactivates CD276 expression. They demonstrate that inhibition of CD276, either through genetic approaches (such as shRNA knockdown, i.e. sh-B7-H3) or blocking antibody (aB7-H3), leads to delayed growth of mTORC1-hyperactive (TSC2-deficient) tumor cells in immunocompetent mice. The increased immunogenicity of B7-H3 knockdown, TSC2-deficient tumors is associated with improved T cell tumor infiltration, T cell cytokine production-IFN γ in particular, and tumor cell response to IFN γ signaling. Finally, they show that B7-H3 inhibition increases the presence of intratumoral cytolytic CD38+CD39+CD4+ T cells, which are also found increased in RCC patients responsive to anti-PD1 checkpoint blockade therapy (ICB).

There are many strengths in this study. aB7-H3 therapy is already in clinical trials. A better understanding of B7-H3 biology clearly has direct clinical relevance to cancer immunotherapy. The study was well-designed, the experiments were technically sound and well-executed. The data presented are extensive and particularly insightful in revealing the molecular mechanisms regarding mTORC1 regulation of B7-H3. However, some aspects of the study need to be clarified and strengthened to increase the rigor and impact of the study.

Specific questions related to data presented:

1. The subtitle “B7-H3 expression and mTORC1 signatures associate with prognosis and tumor immune infiltrate” appears to be ambiguous without spelling out the positive or negative correlation. It should be more specific by stating the nature of the association.

In Fig 1, mTORC1 scores (Fig. 1b) do not always correlate with CD276 scores (Fig. 1d) in terms of clinical come. For example, CD276 scores are associated with worse clinical outcome in MESO and KIPAN but mTORC1 scores have no such association. The authors may comment on whether B7-H3 can also be regulated via mTORC1-independent mechanism.

It is not obvious why the authors stated that tumors with high CD276 scores (Fig. 1j) fall in the C3-inflammatory category.

Lines 148-149, Fig. 1m should be Fig. 1l, Fig. 1n should be Fig. 1m.

2. In Fig. 5b, the tumor sizes for sh-B7-H3 (1) and sh-B7-H3 (2) tumor were still small by day 55, and mice are expected to live beyond 55 days. However, the mouse survival curve in Fig. 5c shows that all mice died by day 55. Was this due to arbitrary termination of the experiment? If by day 55 the B7-H3 KO tumor-bearing mice were still alive, the survival curves for these mice should be corrected.

Compared to sh-B7-H3 knockdown, B7-H3 inhibition by aB7-H3 ab injection had a modest beneficial effect (Extended data Fig. 3i-j). Was this due to the quality of the antibody or other factors? The justification of the use of aB7-H3 ab in TSC2+/- mice should be briefly mentioned in the text. Do TSC2+/- mice develop nonmalignant renal tumors? In these mice, nontumor cells may also be mTORC-hyperactive and thus can be the target of aB7-H3 Ab.

3. The section under “Intact host IFN- γ , tumor cell IFN- γ signaling, and MHC-II expression are necessary to achieve a positive response to B7-H3 inhibition” does not provide any data that correlate tumor cell MHC-II expression to the observed positive response to B7-H3 inhibition (sh-B7-H3). Extended Data Fig. 5h-l show that MHC-II expression in tumor cells was dictated by the presence of IFN γ regardless of the status of B7-H3. The figure legend for Extended Data Fig. 5h-l should specify the use of IFN γ . Knockdown of CIITA in sh-B7-H3 tumor cells may help determine the importance of MHC-II in the experimental system.

4. For Fig. 6h-l, when were CD8 and CD4 T cells isolated from tumor-bearing mice for cytokine ICS? It is striking that T cells from sh-NC tumor-bearing mice were unable to produce IFN γ /TNF α at all upon PMA/inomycin stimulation while T cells from sh-B7-H3-bearing mice were polyfunctional. Does this result suggest that B7-H3 on tumor cells causes severe T cell exhaustion? Was there difference in IFN γ in the serum/plasma from mice bearing sh-NC vs ah-B7-H3 tumors?

Data shown in Fig. 6j-k should include isotype controls to justify the gating. The pattern of IFN γ ICS of CD4 T cells looked unconvincing, and the difference in CD4 IFN γ production between the groups was minor (2% vs 4%). Naïve T cell cytokine profile upon PMA/inonmycin stimulation in the absence of tumor cells should be included for comparison. Overall, data in Fig. 6j-k seem to be weak and a distraction.

5. The subtitle “Intratumoral CD4+ T cells are required for B7-H3’s immunosuppressive effects in Tsc2-null tumors” is a bit confusing. The data presented in section suggest that CD4 T cells play an important role in controlling the growth of B7-H3-deficient tumors. It’s not obvious what the “B7-H3’s immunosuppressive effects” refer to.

In studies described in Fig. 7a-c, why CD4 or CD8 depletion antibodies were injected to mice so

frequently? Most studies inject depleting Ab once or twice a week.

The results from CD8-depletion (Fig. 7c) and CD8KO mice (Extended data Fig. 6e) both suggest that CD8 T cells contributed to inhibition of sh-B7-H3 tumor growth, though in a role less prominent as CD4 T cells. The authors did not address which cell subset, CD4 or CD8 or both, is the major source of IFN γ that triggered IFN γ signaling in tumor cells.

The identification of cytolytic CD38+CD39+CD4 T cells is quite interesting, but the manuscript runs short of proving the relevance of these cells to the observed better growth control of B7-H3-deficient tumors. Are these cytolytic CD4 T cells specialized in killing MHCII+ tumor cells or they also provide IFN γ to tumor cells? An in vitro study to demonstrate the killing capability of these cytolytic CD4 T cells would minimally support their presumed role in vivo.

6. One clinically relevant question is how B7-H3 expression is related to PD-L1 expression. It is controversial whether mTORC1 positively or negatively regulates PD-L1. The authors should examine the PD-L1 levels in the tumor cells used in this study and explain whether the status of PD-L1 has anything to do with the observed outcomes. It will be interesting to know whether the increased IFN γ production from T cells in sh-B7-H3-bearing mice would lead to increased PD-L1 expression in tumor cells, which may cause T cell dysfunction and outgrowth of B7-H3-deficient tumors eventually. Do the high levels of PD1 in T cells (both CD8 and CD4) in mice with B7-H3-deficient tumors (Fig. 7 g and i) provide the rationale for combining B7-H3 blockade with PD1/PD-L1 blockade therapy?

Reviewer #3 (Remarks to the Author): with expertise in mTOR signaling, cancer

Heng-Jia Liu et al. present a work describing the contribution of B7-H3/CD276 in tumor cells towards the limitation of tumoral T-cell infiltration, thus driving tumor immune evasion. The authors revealed that the mTORC1/S6K pathway positively regulates B7-H3. They conclude that the inhibition of this protein leads to an increase in the number of cytotoxic CD38+ CD39+ CD4+ T cells inside the tumor.

The manuscript is well constructed and the experiments were well conducted. To get to the conclusions, the authors performed experiments in culture tumor cells and xenograft models. Different approaches were used, including bioinformatic studies, mass cytometry, CITE-seq, ChIP-qPCR, in vitro phosphorylation and ubiquitination assays.

Still, a number of questions require additional work to be clarified. The following questions are presented with no particular order of priority.

1. Figure 1G. In the text (lines 133-135), the authors claim that they use “130 tumors representing 12 different tumor types” to describe the correlation between protein expression of B7-H3 and phospho-S6. It might be appropriate that they describe which are these 12 tumor types. In this same figure, they use phospho-S6 as a marker of mTORC1 activation but, as it is not a direct target of this complex, it is recommended that they use a direct target such as S6-kinase.

2. Figure 2A-C. Phospho-S6 is used to monitor mTORC1 activation. It is recommended to use a direct target of mTORC1 (S6 is not), such as S6-kinase, to follow mTORC1 activation, as these authors did in other figures of the paper (Fig. 2M).

3. For every WB panel using anti-phospho antibodies, the total antibody counterpart for the specific protein should be included. These controls are missing in a number of figures (Figs 2G, 2H, 2I, 4A, 4B, 4F).

4. In the PDF file, Figs 2 is not shown completely, which precludes its evaluation.

5. Figure 3. The authors employed TSC2 KO MEFs to show the downregulation of Cd276 expression upon YY2 silencing. Does YY2 overexpression restore Cd276 expression in rapamycin treated TSC KO cells? This will help to mechanistically locate YY2 in the mTORC1-Cd276 connection.

6. Figure 4H-K. Does MG132 treatment prevent YY2 degradation upon mTORC1/S6K inhibition?

7. Figure 6D. The authors describe that interferon gamma (IFN- γ) pathway is upregulated upon B7-H3 silencing. As IFN- γ enhances the expression of MHC-I and MHC-II, the authors pertinently analysed the expression levels of MHC-II in sh-B7-H3 mice. What about MHC-I?

8. Figure 6E. The conclusion that nuclear localization of STAT1 is increased upon Cd276 silencing is not well sustained by the IF images, which do not clearly show such a localization. Either increasing IF images resolution or using alternative approaches (cellular fractionation) is required to clarify this point.

Reviewer #4 (Remarks to the Author): with expertise in cancer immunology, omics

In this manuscript, Liu and colleagues present compelling data suggesting mTORC1 activity is sufficient to maintain CD276 expression in tumor cells through indirect activation of the YY2 transcription factor. Subsequently, they show that a loss of CD276 abrogates tumor growth in-vivo due to a loss in immune evasion characteristics related to increased T cell infiltration, increased MHC-II expression, and activation of CD4 (and to some degree CD8) T cells within the microenvironment. This study includes very thorough mechanistic experimentation and, for the most part, appears relevant to a broad spectrum of cancers. Discussed below are specific points and suggestions for the authors to consider:

A bulk of my comments pertain to figure 1, which is the only part of the manuscript I have found problematic. To be clear, this is a very thorough manuscript, and I think the study overall was well done. The first figure, however, detracts from the overall story presented in figures 2-7. My comments:

1. The manuscript is very convincing in the connection between mTOR activity and CD276 expression; however, the introduction and first figure have trouble convincing me of the prior justification for their connection. Until the two are explicitly tested in Figure 2, it is unclear if there is a justifiable reason to believe the two are connected.

2. In figure 1 and lines 122-132, does expression evenly distribute amongst the cancers? It could be that those diagnoses with much longer survival times are also more likely to have low basal mTORC1 activity or low CD276. As far as the methods described, the high-low cutpoint was determined from all samples pooled. This could significantly confound the survival analysis done in figure 1. Given this facet, the hazard ratios are more convincing of the point, except many of the TCGA subtypes are in such small quantities that subsetting, as described, would yield a comparison between 8-12 samples in each group (high/low). Additionally, mixing cancers that have different hazards in unequal proportions means unequal hazards, which makes log-rank tests fail. I would recommend confirming that proportional hazards are met and that a minimum sample number is present before including the HR's. Finally, the sheer number of log-rank tests likely merit accounting for multiple comparisons. a. Overall, the author claims this is evidence of "pan-cancer significance". In reality, it is more subtype-specific significance.

3. Much of the article utilizes a mouse kidney cancer model cell line, and yet none of the TCGA kidney cancer subtypes are included. This is a missed opportunity to support the work later in the paper. It is a little suspicious that it is not included, considering the large number of them in TCGA. Likewise, one must ask why GBM and LGG but no other subtypes are combined similarly.

4. Line 135 and Figure 1H, the correlation is not very convincing that there is a "strong" correlation. P-values must not be confused with r values. The significance may suggest it's not a random correlation, but an r value of 0.36 is a fairly weak correlation, considering both scores are scaled the same and have fairly consistent distribution. In fact, the authors used a spearman test, so a rho should be displayed so as not to confuse the correlation with a Pearson r. Despite this, there are a

significant number of ties in this data, and spearman notoriously does not handle ties well. This likely inflates the spearman rho displayed, and a Pearson correlation is likely even lower. Spearman is likely not appropriate here because of the significant number of ties, and both variables are scaled the same and are distributed equally.

5. Much like the above comment, figures l and m are also likely overinterpretations. There are so many datapoints that any correlation value is likely to be significant. However, all but the last one suggest there is no amount of x that determines y.

6. The conclusions stated in lines 141-144 are not supported by the data displayed in the figure. Nearly all of the distributions in I and J are identical except for C5 CD276 expression. Likewise for Figure 1k, where there is no mention of how one box for each "Type"/"Cell Type" pair is obtained, considering many samples were tested.

The rest of my comments on the remaining parts of the manuscript are as follows:

1. Throughout the manuscript, 105K cells deficient in Tsc2 are used as a cancer model with hyperactive mTOR. It's clear this is a very stable model, but I am curious how the results hold in a more natural context of mTOR activation (i.e., growth factors or amino acid supplement) or under a more direct activation of mTORC1 like site mutagenesis for constitutive activation? Much of the paper is devoted to comparing Tsc2 KD to baseline, but overexpression or direct constitutive activation is more direct and recapitulate what happens in cancer.

2. There is no mention of how JASPAR was used. What were the inputs, species, approach, dataset used, etc. As far as I can find, JASPAR only lists the CCAT motif for humans.

3. In Figures 6d and 6f, MHC-II expression measured by lysate and IHC isn't entirely fair for evaluating antigen presentation. Extended data figure 5h essentially shows that the surface expression of MHC-II doesn't change. It also cannot be ignored that mTOR itself can induce antigen presentation.

4. In all in-vivo tumor models, except for one (Extended Figure 3i), B7-H3 is either deleted before implantation or anti-B7-H3 treatment starts at the time of implantation. It would be useful to see more examples of B7-H3 inhibition in established tumors. The effectiveness of inhibition is expected to be most effective when the treatment is prophylactic, and the tumor hasn't been infiltrated by the immune system. On a related note, having an orthotopic model (either genetic or implantable) would show more breadth.

5. Without overinterpreting a UMAP, it is strange to see naive CD4 and CD8 T cells segregate so well, and yet the CD4 and CD8 cytotoxic T cell populations are directly adjacent to each other. Given that samples were processed at the same time and pooled, it may not have been necessary to integrate the data as it can overcorrect. Have you examined the data without integration to see if it is merited?

6. Figure 7p, the signature used on TCGA data is derived from scRNAseq of sorted T cells. Since tumor cells were not included in the process to generate the signature, the resultant genes could be present or aberrant in tumor cells. When calculating bulk tumor samples, the tumor cells themselves could affect the scoring independent of the actual cell quantities the authors wish to measure. It may be useful to take a ConsensusTME approach and remove genes that have a $r > 0.25$ correlation with tumor purity before testing the score.

We are very appreciative of the Reviewers for their support and assistance in strengthening our manuscript for publication. As suggested, we have performed additional experiments to confirm and validate our findings. The revised manuscript contains 23 new/revised figure panels (Figures 1d, 1e, 1f, 1l, 1j, 1k, 1l, 2a, 2b, 2c, 2g, 2h, 2i, 4q, 6g, 6h, 6l, 6m, 7p: Extended figures s1k, s1l, s3o, s7f). We have included figure panels in this rebuttal letter to further clarify our findings. In addition, we incorporated additional references and text to provide a more complete interpretation of the data.

Our point-by-point responses are below.

Reviewer #1 (Remarks to the Author):

1. The paper “Tumor-expressed B7-H3 mediates the inhibition of antitumor T-cell functions in ovarian cancer insensitive to PD-1 blockade therapy” already showed that B7-H3 expressed on tumor cells, but not host cells, had a dominant role in suppressing antitumor immunity, which was dependent on CD8⁺ T cells.

Response: We thank the reviewer for pointing out this publication, in which the authors found that B7-H3 expression on tumor cells and tumor-infiltrating antigen presenting cells correlates with T-cell exhaustion in ovarian cancer patients. Inhibition of B7-H3 in their models inhibits the function of CD8⁺ T cells. We have added this citation to our introduction (line 87). These data support our results, although we note that we have taken many additional steps, including B7-H3 expression is regulated by mTORC1/S6K via YY2, intact host INF- γ , tumor cell INF- γ signaling via STAT1, and MHC-II expression are required to achieve a positive response to B7-H3 inhibition therapy. Furthermore, we have shown for the first time CD4⁺ T cells (in addition to CD8⁺ T cells) are critical for B7-H3 inhibition therapy, our CITE-seq data revealed that inhibition of B7-H3 promotes the accumulation of cytotoxic CD4⁺ T cells that expresses CD38 and CD39.

2. The figures are not well prepared. For example, Figure 2f and Figure 2l are missing? Figure 1m should be figure 1l and Figure 1n is Figure 1m. In Figure 2n, what does each color represent? Extended Data Fig. 3c, d did not label cell type. Extended Data Fig. 7f cluster annotations of CD4-CP and CD4-GZMK were wrong in the color.

Response: We apologize for the missing Figure 2f and Figure 2l, which are included in the revised manuscript. We have also fixed the errors related to Figures 1m and 1n. In Figure 2n, the color codes (Black = Ctrl siRNA, Red = Raptor siRNA, Orange = Rictor siRNA, Green = mTOR siRNA, Blue = S6K siRNA, Purple= 4EBP1 siRNA) are now included and again we apologize for the omission in the original version of the manuscript. Cell types are also now labeled in the revised Extended Data Fig. 3c. The colors for the cluster annotations for CD4-CP and CD4-GZMK in Extended Data Fig. 7f have been corrected. We are grateful to the Reviewer for picking up these errors and omissions.

3. Most tumor cells are Tsc2-wild type, and B7-H3 protein is also expressed in Tsc2 wild-

type MEFs (Fig. 2m). The authors should at least discuss the regulator of B7-H3 expression in TSC2-WT tumor.

Response: We agree that this is an important point in terms of the impact of our work on other tumor types that have wild-type TSC2. mTOR is aberrantly overactivated in more than 70% of human tumors through multiple mechanisms which involve the positive and negative regulators of the mTOR pathway, including phosphoinositide 3-kinase (PI3K)/Akt, mitogen-activated protein kinase (MAPK), vascular endothelial growth factor (VEGF), nuclear factor- κ B, and p53, etc.,

To determine whether our results are applicable to tumors with wild-type TSC2, we treated A549, T47D, and PC3 cells with rapamycin and Torin 1 (Extended data Figure 1a-c). The mRNA and protein expression of B7-H3 is suppressed in all three cell lines by rapamycin and Torin 1, indicating that mTORC1 regulates B7-H3 expression in cells with wild-type TSC2.

4. B7-H3 expression is lower in sh-B7H3(2) group than that in sh-B7H3(1) group (Fig.6d), however, the tumor volume in sh-B7H3(2) group is bigger than that in sh-B7H3(1) group (Fig.5b) in WT mice and the percentage of CD4⁺ T cells is lower in sh-B7H3(2) group than that in sh-B7H3(1) group (Fig.5i). The authors should also analyze the correlation between B7-H3 knockdown efficiency with tumor growth.

Response: We apologize that sh-B7-H3 (1) and sh-B7-H3 (2) in Figure 6d were incorrectly labeled; this has been corrected in the revised manuscript. The knockdown efficiency of sh-B7-H3 (1) is actually greater than sh-B7-H3 (2), as indicated in extended data Figure 3a. Consistent with this, tumor volume is greater in the sh-B7-H3 (2) group than in the sh-B7-H3 (1), as shown in Figure 5d, and the percentage of CD4⁺ T cells is lower in sh-B7-H3 (2), as shown in Fig. 5i. We again apologize for the confusion created by the incorrect labeling in the original version of Figure 6d.

Reviewer #2 (Remarks to the Author):

Liu et al reported in this manuscript that mTORC1 critically controls tumor cell expression of B7-H3/CD276, which functions as a co-inhibitory molecule to suppress antitumor T cell response and evade immunosurveillance. They show that mTORC1 phosphorylates the transcriptional factor YY2 via S6K, preventing degradation of YY2 from Smurf1-mediated ubiquitination. YY2 binds to CD276 promoter and transactivates CD276 expression. They demonstrate that inhibition of CD276, either through genetic approaches (such as shRNA knockdown, i.e. sh-B7-H3) or blocking antibody (aB7-H3), leads to delayed growth of mTORC1-hyperactive (TSC2-deficient) tumor cells in immunocompetent mice. The increased immunogenicity of B7-H3 knockdown, TSC2-deficient tumors is associated with improved T cell tumor infiltration, T cell cytokine production-IFN γ in particular, and tumor cell response to IFN γ signaling. Finally, they show that B7-H3 inhibition increases the presence of intratumoral cytolytic CD38⁺CD39⁺CD4⁺ T cells, which are also found increased in RCC patients responsive to anti-PD1 checkpoint blockade therapy (ICB).

There are many strengths in this study. aB7-H3 therapy is already in clinical trials. A better understanding of B7-H3 biology clearly has direct clinical relevance to cancer immunotherapy. The study was well-designed, the experiments were technically sound and well-executed. The data presented are extensive and particularly insightful in revealing the molecular mechanisms regarding mTORC1 regulation of B7-H3. However, some aspects of the study need to be clarified and strengthened to increase the rigor and impact of the study.

Response: We thank the Reviewer for the many positive comments about the strengths of this work, including the clinical relevance, the design of the study, the technically well-executed experiments, and the data presentation.

Specific questions related to data presented:

1. The subtitle “B7-H3 expression and mTORC1 signatures associate with prognosis and tumor immune infiltrate” appears to be ambiguous without spelling out the positive or negative correlation. It should be more specific by stating the nature of the association.

Response: We have changed the subtitle to “High B7-H3 expression and high mTORC1 signatures correlate with poorer prognosis and less anti-tumor immune cell infiltrates.”

In Fig 1, mTORC1 scores (Fig. 1b) do not always correlate with CD276 scores (Fig. 1d) in terms of clinical come. For example, CD276 scores are associated with worse clinical outcome in MESO and KIPAN but mTORC1 scores have no such association.

Response: We thank the Reviewer for pointing this out. According to Reviewer's 4 suggestion, we have performed this analysis again using the multivariate COX proportional hazards model. In this new analysis, we selected the top 11 cancer types with death events greater than 20. We found that the mTORC1 hazard ratio score always correlates with the CD276 mTORC1 hazard ratio score (Figs 1b, d, f).

The authors may comment on whether B7-H3 can also be regulated via mTORC1-independent mechanism.

Response: The Reviewer is correct that B7-H3 can be regulated via several mechanisms, including microRNAs, immunoglobulin-like transcript (ILT) 4, and autophagy, and we have added a comment about these alternative mechanisms to the Introduction (line 97-99). Whether these mechanisms are truly mTORC1-independent is not yet fully understood.

It is not obvious why the authors stated that tumors with high CD276 scores (Fig. 1j) fall in the C3-inflammatory category.

Response: We apologize for this oversight. We removed this sentence from the Results section and replaced it with "high CD276 fall in the C6-TGF-β dominant category".

Lines 148-149, Fig. 1m should be Fig. 1l, Fig. 1n should be Fig. 1m.

Response: We thank the Reviewer for pointing this out. These errors are now corrected.

2. In Fig. 5b, the tumor sizes for sh-B7-H3 (1) and sh-B7-H3 (2) tumor were still small by day 55, and mice are expected to live beyond 55 days. However, the mouse survival curve in Fig. 5c shows that all mice died by day 55. Was this due to arbitrary termination of the experiment? If by day 55 the B7-H3 KO tumor-bearing mice were still alive, the survival curves for these mice should be corrected.

Response: We apologize that this was not clear. Figure 5b shows tumor-free survival, not overall survival. When a tumor becomes palpable (~100 mm³), the mouse is designated as no-longer tumor-free. We have now made this clearer in the Methods section (line 881-882).

Compared to sh-B7-H3 knockdown, B7-H3 inhibition by aB7-H3 ab injection had a modest beneficial effect (Extended data Fig. 3i-j). Was this due to the quality of the antibody or other factors? The justification of the use of aB7-H3 ab in Tsc2^{+/-} mice should be briefly mentioned in the text. Do Tsc2^{+/-} mice develop nonmalignant renal tumors? In these mice, nontumor cells may also be mTORC-hyperactive and thus can be the target of aB7-H3 Ab.

Response: This clone of the B7-H3 neutralizing antibody (MJ18) used in our Tsc2^{+/-} mice was purchased commercially from Bio X cell. It is difficult to know why the beneficial effects of the antibody were less robust than B7-H3 knockdown.

Regarding the tumors that develop in the Tsc2^{+/-} mice, these are primarily nonmalignant. Several groups, including ours, have shown that the nontumor cells are not mTORC1 hyperactive¹⁻⁴, consistent with the requirement for a “second hit” mutation in the wild-type copy of TSC2.

3. The section under “Intact host IFN- γ , tumor cell IFN- γ signaling, and MHC-II expression are necessary to achieve a positive response to B7-H3 inhibition” does not provide any data that correlate tumor cell MHC-II expression to the observed positive response to B7-H3 inhibition (sh-B7-H3). Extended Data Fig. 5h-I show that MHC-II expression in tumor cells was dictated by the presence of IFN γ regardless of the status of B7-H3. The figure legend for Extended Data Fig. 5h-I should specify the use of IFN γ . Knockdown of CIITA in sh-B7-H3 tumor cells may help determine the importance of MHC-II in the experimental system.

Response: We thank the reviewer for these comments and apologize for not making clear the connection of tumor cell MHC-II expression to the observed positive response to B7-H3 inhibition. We have now performed additional genetic studies with CRISPR-Cas9 deletion of CIITA in Tsc2-deficient cells with sh-B7-H3 and examined the *in vivo* growth of these cells. These studies support our conclusion that deletion of CIITA results in decreased MHC-II expression leading to increased tumor growth in shB7-H3 cells compared to controls (Fig. 6h).

In addition, we have specified the use of IFN γ in the figure legend for Extended Data Fig. 5h-l.

4. For Fig. 6h-l, when were CD8 and CD4 T cells isolated from tumor-bearing mice for cytokine ICS? It is striking that T cells from sh-NC tumor-bearing mice were unable to produce IFN γ /TNF α at all upon PMA/inomycin stimulation while T cells from sh-B7-H3-bearing mice were polyfunctional. Does this result suggest that B7-H3 on tumor cells causes severe T cell exhaustion? Was there difference in IFN γ in the serum/plasma from mice bearing sh-NC vs sh-B7-H3 tumors?

Response: Our experimental results in Fig. 6h-l used CD8 and CD4 T cells isolated from tumor-bearing mice about a month after inoculation. Our CITE-seq data suggest that B7-H3 on tumor cells contributes to T cell exhaustion, since both CD8 $^+$ and CD4 $^+$ T cells from sh-B7-H3 tumors have expression of IFN γ , TNF α , and Gzmb (Figs. 7i and 7g).

We isolated serum from sh-NC and sh-B7-H3 mice and used a commercial kit (BioLegend)

to measure serum IFN γ by ELISA. undetectable in both groups (see Fig. difference in IFN γ only occurs in the

We found that serum IFN γ is below). It is possible that the tumor milieu.

Data shown in Fig. 6j-k (now called Fig. 6l-m) should include isotype controls to justify the gating. The pattern of IFN γ ICS of CD4 T cells looked unconvincing, and the difference in CD4 IFN γ production between the groups was minor (2% vs 4%). Naïve T cell cytokine profile upon PMA/inomycin stimulation in the absence of tumor cells should be included for comparison. Overall, data in Fig. 6j-k seem to be weak and a distraction.

Response: We have included the isotype controls and Naïve T cell cytokine upon PMA/inomycin stimulation in the absence of tumor cells profile for Fig 6j-k (now called Fig. 6l-m) . The pattern of IFN γ (single positive) ICS of CD4⁺ T cells shows a nearly three-fold difference (6% vs. 17%). The 2% vs. 4% difference is in the double positive IFN γ /TNF α -producing CD4⁺ T cells. We have kept these data in the revision, since we believe they are important, but can remove them if the Reviewer and/or Editor prefer.

5. The subtitle “Intratumoral CD4+ T cells are required for B7-H3’s immunosuppressive effects in Tsc2-null tumors” is a bit confusing. The data presented in section suggest that CD4 T cells play an important role in controlling the growth of B7-H3-deficient tumors. It’s not obvious what the “B7-H3’s immunosuppressive effects” refer to.

Response: We appreciate these suggestions and apologize for the oversight. We have changed the subtitle to “Intratumoral CD4⁺ T cells are required for the impact of B7-H3 knockdown on tumor growth.”

In studies described in Fig. 7a-c, why CD4 or CD8 depletion antibodies were injected to mice so frequently? Most studies inject depleting Ab once or twice a week.

Response: We depleted CD4⁺ and CD8⁺ T cells once every three days, which is about twice a week (Fig. 7a). It is possible that similar levels of depletion could be achieved with once-a-week injections.

The results from CD8-depletion (Fig. 7c) and CD8KO mice (Extended data Fig. 6e) both suggest that CD8 T cells contributed to inhibition of sh-B7-H3 tumor growth, though in a role less prominent as CD4 T cells. The authors did not address which cell subset, CD4 or CD8 or both, is the major source of IFN γ that triggered IFN γ signaling in tumor cells.

Response: We data showed that CD8⁺ T cells inhibit sh-B7-H3 tumor growth less prominently than CD4⁺ T cells. Our CITE-seq data (Fig. 7I) indicate that effector CD4⁺ T cells expressing Gzmk are the dominant IFN γ -producing population.

The identification of cytolytic CD38⁺CD39⁺CD4⁺ T cells is quite interesting, but the manuscript runs short of proving the relevance of these cells to the observed better growth control of B7-H3-deficient tumors. Are these cytolytic CD4⁺ T cells specialized in killing MHCII⁺ tumor cells or they also provide IFN γ to tumor cells? An in vitro study to demonstrate the killing capability of these cytolytic CD4⁺ T cells would minimally support their presumed role in vivo.

Response: We thank the reviewer for this critical suggestion and performed ex vivo studies to directly assess the killing capability of these CD38⁺CD39⁺CD4⁺ T cells. CD4⁺ TILs (CD45⁺CD3⁺CD4⁺CD25⁻CD38⁺CD39⁺) were isolated by FACS from single-cell suspensions generated from Tsc2-deficient 105K tumors. CD38⁺CD39⁺CD25⁻CD4⁺ TILs were activated by Dynabeads Mouse T-Activator CD3/CD28 for 2 weeks and co-cultured with Tsc2-deficient 105K cells pretreated with 10ng/ml of mouse IFN γ in order to upregulate MHC-II. We have also performed these experiments with an MCH-II blocking antibody, looking at the impact on the killing ability of CD38⁺CD39⁺CD25⁻CD4⁺ TILs. We found CD38⁺CD39⁺CD25⁻CD4⁺ TILs triggered increased tumor cell death. Pretreating tumor cells with MHC-II blocking antibody inhibited CD38⁺CD39⁺CD25⁻CD4⁺ TIL-mediated tumor cell death (Fig. 7p). We conclude that these CD38⁺CD39⁺CD25⁻CD4⁺ TILs are specialized in killing MHC-II⁺ tumor cells and producing IFN γ .

7p

6. One clinically relevant question is how B7-H3 expression is related to PD-L1 expression. It is controversial whether mTORC1 positively or negatively regulates PD-L1. The authors should examine the PD-L1 levels in the tumor cells used in this study and explain whether the status of PD-L1 has anything to do with the observed outcomes. It will be interesting to know whether the increased IFN γ production from T cells in sh-B7-H3-bearing mice would lead to increased PD-L1 expression in tumor cells, which may cause T cell dysfunction and outgrowth of B7-H3-deficient tumors eventually. Do the high levels of PD1 in T cells (both CD8 and CD4) in mice with B7-H3-deficient tumors (Fig. 7 g and i) provide the rationale for combining B7-H3 blockade with PD1/PD-L1 blockade therapy?

Response: We agree that the relationship between B7-H3 expression and PD-L1 expression may be clinically relevant, and we also agree that it is unclear whether mTORC1 positively or negatively regulates PD-L1. Multiple data sets indicate that PD-L1 is not upregulated at baseline in TSC-associated tumors, including our prior work using immunohistochemistry ⁵ and our recent work using single cell RNA sequencing ⁶. Interestingly, in a new analysis, we examined PD-L1 (CD274) expression in our ex vivo sorted tumor cells and found that PD-L1 (CD274) is upregulated by 2.56-fold (padj = 1.46E-10) in sh-B7-H3 tumor cells compared to sh-NC (Extended Data Table 1). These data could provide a rationale for combining B7-H3 blockade with PD-1/PD-L1 blockade therapy.

Reviewer #3 (Remarks to the Author):

Heng-Jia Liu et al. present a work describing the contribution of B7-H3/CD276 in tumor cells towards the limitation of tumoral T-cell infiltration, thus driving tumor immune evasion. The authors revealed that the mTORC1/S6K pathway positively regulates B7-H3. They conclude that the inhibition of this protein leads to an increase in the number of cytotoxic CD38⁺ CD39⁺ CD4⁺ T cells inside the tumor.

The manuscript is well constructed and the experiments were well conducted. To get to the conclusions, the authors performed experiments in culture tumor cells and xenograft models. Different approaches were used, including bioinformatic studies, mass cytometry, CITE-seq, CHIP-qPCR, in vitro phosphorylation and ubiquitination assays.

We thank the Reviewer for the positive comments about the construction of the manuscript and the diversity of experimental approaches.

Still, a number of questions require additional work to be clarified. The following questions are presented with no particular order of priority.

1. Figure 1G. In the text (lines 133-135), the authors claim that they use “130 tumors representing 12 different tumor types” to describe the correlation between protein expression of B7-H3 and phospho-S6. It might be appropriate that they describe which are these 12 tumor types. In this same figure, they use phospho-S6 as a marker of mTORC1 activation but, as it is not a direct target of this complex, it is recommended that they use a direct target such as S6-kinase.

Response: We thank the reviewer for this suggestion and apologize for the missing tumor-type details in the legend; this is now corrected. As suggested, we performed immunohistochemical staining on the same cohort of patient samples using an anti-phosphorylated S6-kinase (T389) antibody and scored its correlation with B7-H3 expression. We saw a significant correlation of phosphorylated S6-kinase with B7-H3 (Figs. 1i & 1j).

2. Figure 2A-C. Phospho-S6 is used to monitor mTORC1 activation. It is recommended to use a direct target of mTORC1 (S6 is not), such as S6-kinase, to follow mTORC1 activation, as these authors did in other figures of the paper (Fig. 2M).

Response: As suggested, we have now included both phospho S6-Kinase and total S6-kinase in Figure 2A-C, and found that it parallels the phospho-S6 results.

3. For every WB panel using anti-phospho antibodies, the total antibody counterpart for the specific protein should be included. These controls are missing in a number of figures (Figs 2G, 2H, 2I, 4A, 4B, 4F).

Response: We have now included the total antibody counterpart for Figures 2G, H, I, 4A, 4B, and 4F.

4. In the PDF file, Figs 2 is not shown completely, which precludes its evaluation.

Response: We apologize for this issue and are not certain why the PDF file was incomplete. We will work with the Editors to ensure that this does not happen again.

5. Figure 3. The authors employed TSC2 KO MEFs to show the downregulation of Cd276 expression upon YY2 silencing. Does YY2 overexpression restore Cd276 expression in rapamycin treated TSC KO cells? This will help to mechanistically locate YY2 in the mTORC1-Cd276 connection.

Response: We thank the Reviewer for this excellent suggestion. We performed new experiments in which we overexpressed Myc-YY2 in Tsc2-deficient 105K cells and examined CD276/B7-H3 protein expression after 24 hours of rapamycin treatment. As the Reviewer predicted, YY2 overexpression partially restores B7-H3 expression in the rapamycin-treated cells (Fig. 4q).

6. Figure 4H-K. Does MG132 treatment prevent YY2 degradation upon mTORC1/S6K inhibition?

Response: As suggested, we treated cells with MG132, and found that this prevents GFP-YY2 degradation upon S6K inhibition (Fig. 4h).

7. Figure 6D. The authors describe that interferon gamma (IFN- γ) pathway is upregulated upon B7-H3 silencing. As IFN- γ enhances the expression of MHC-I and MHC-II, the

authors pertinently analyzed the expression levels of MHC-II in sh-B7-H3 mice. What about MHC-I?

Response: Several MHC-I genes are also upregulated in sh-B7-H3 tumor cells compared to sh-NC. We added these genes to the heatmap in Fig. 6c.

8. Figure 6E. The conclusion that nuclear localization of STAT1 is increased upon Cd276 silencing is not well sustained by the IF images, which do not clearly show such a localization. Either increasing IF images resolution or using alternative show approaches (cellular fractionation) is required to clarify this point.

Response: We have included higher magnification images (Fig. 6e) to show more clearly that STAT1 is localized to the nucleus.

Reviewer #4 (Remarks to the Author):

In this manuscript, Liu and colleagues present compelling data suggesting mTORC1 activity is sufficient to maintain CD276 expression in tumor cells through indirect activation of the YY2 transcription factor. Subsequently, they show that a loss of CD276 abrogates tumor growth in-vivo due to a loss in immune evasion characteristics related to increased T cell infiltration, increased MHC-II expression, and activation of CD4 (and to some degree CD8) T cells within the microenvironment. This study includes very thorough mechanistic experimentation and, for the most part, appears relevant to a broad spectrum of cancers. Discussed below are specific points and suggestions for the authors to consider:

A bulk of my comments pertain to figure 1, which is the only part of the manuscript I have found problematic. To be clear, this is a very thorough manuscript, and I think the study overall was well done. The first figure, however, detracts from the overall story presented in figures 2-7. My comments:

1. The manuscript is very convincing in the connection between mTOR activity and CD276 expression; however, the introduction and first figure have trouble convincing me of the prior justification for their connection. Until the two are explicitly tested in Figure 2, it is unclear if there is a justifiable reason to believe the two are connected.

Response: We thank the Reviewer for the positive comments about the compelling data, the thorough mechanistic experimentation, and the potential for relevance to a spectrum of cancers. As detailed below, we have worked hard to further connect Fig. 1 with the other data in the manuscript.

2. In figure 1 and lines 122-132, does expression evenly distribute amongst the cancers? It could be that those diagnoses with much longer survival times are also more likely to have low basal mTORC1 activity or low CD276. As far as the methods described, the high-low cutpoint was determined from all samples pooled. This could significantly confound the survival analysis done in figure 1. Given this facet, the hazard ratios are more convincing of the point, except many of the TCGA subtypes are in such small quantities that subsetting, as described, would yield a comparison between 8-12 samples in each group (high/low). Additionally, mixing cancers that have different hazards in unequal proportions means unequal hazards, which makes log-rank tests fail. I would recommend confirming that proportional hazards are met and that a minimum sample number is present before including the HR's. Finally, the sheer number of log-rank tests likely merit accounting for multiple comparisons.

a. Overall, the author claims this is evidence of "pan-cancer significance". In reality, it is more subtype-specific significance.

Response: We apologize for not being clearer about how we processed the samples. We did not pool the samples together before setting the high-low cut point. Instead, we set the high-low cut point for each of the TCGA cancer types separately, and then pooled

together for further analyses of survival. We clarified this in the Methods section (line 984-985).

We thank the reviewer for pointing out that some of the TCGA subtypes are in small quantities and the possible inaccuracy of different hazards in unequal proportions may mean unequal hazards. As suggested, to determine whether there is a connection between survival times and mTOR activity and/or CD276 expression, we reanalyzed the TCGA survival data and selected the top 11 cancers that have death events greater than 20 and stratified the cancer types using the multivariate COX proportional-hazards model. We found that the mTORC1 hazard ratio score always correlates with the CD276 mTORC1 hazard ratio score (Figs 1b, d, f). We agree that it is premature to conclude that there is “pan-cancer significance” and removed this statement from the text.

3. Much of the article utilizes a mouse kidney cancer model cell line, and yet none of the TCGA kidney cancer subtypes are included. This is a missed opportunity to support the work later in the paper. It is a little suspicious that it is not included, considering the large number of them in TCGA. Likewise, one must ask why GBM and LGG but no other subtypes are combined similarly.

Response: We thank the Reviewer for highlighting the connections to kidney cancer, which are extremely interesting to us. The original manuscript included an analysis of the combined TCGA kidney cancer subtypes (abbreviated KIPAN for **K**idney **P**an analysis). After setting the high-low cut off for the three kidney cancer subtypes (clear cell, chromophobe and papillary), only clear cell kidney cancer (KIRC) has death events greater than 20, and it is now included in Fig. 1b, 1d & 1f. As suggested, we separately

reanalyzed GBM and LGG and found that they have a similar hazard ratio trend in the (Fig. 1b, 1d & 1f).

4. Line 135 and Figure 1H, the correlation is not very convincing that there is a “strong” correlation. P-values must not be confused with r values. The significance may suggest it’s not a random correlation, but an r value of 0.36 is a fairly weak correlation, considering both scores are scaled the same and have fairly consistent distribution. In fact, the authors used a spearman test, so a rho should be displayed so as not to confuse the correlation with a Pearson r. Despite this, there are a significant number of ties in this data, and spearman notoriously does not handle ties well. This likely inflates the spearman rho displayed, and a Pearson correlation is likely even lower. Spearman is likely not appropriate here because of the significant number of ties, and both variables are scaled the same and are distributed equally.

Response: We agree that the correlation in Fig 1h should not have been characterized as “strong” and removed the word strong from the Results section. We also revised Fig. 1h and used Pearson’s correlation to indicate the value in Fig 1h.

5. Much like the above comment, figures l and m are also likely overinterpretations. There are so many datapoints that any correlation value is likely to be significant. However, all but the last one suggest there is no amount of x that determines y.

Response: We again apologize for the oversight and agree with the Reviewer that there is a trend that mTORC1 signature and CD276 expression inversely correlate with lymphocyte infiltration (Fig. 1n) and positively correlate with TFG- β response (Fig. 1o). We have corrected the wording in the result section. We have kept these data in the revision, but can remove them if the Reviewer and/or Editor prefer.

6. The conclusions stated in lines 141-144 are not supported by the data displayed in the figure. Nearly all of the distributions in I and J are identical except for C5 CD276 expression. Likewise for Figure 1k, where there is no mention of how one box for each "Type"/"Cell Type" pair is obtained, considering many samples were tested.

Response: We agree with the Reviewer that the major differences in Figs. 1I and 1J (now are Figs. 1k and 1l) relate to the C5 CD276 expression. We have revised our description of this in the Results section to reflect this (line 146-149).

For Fig 1k (now is Fig 1m), we have included details of how one box for each type is obtained in the legend. Briefly, the heatmap indicates the immune-cell mean scores of all patients in high CD276 and high mTORC1 tumors and low CD276 - low mTORC1 tumors.

The rest of my comments on the remaining parts of the manuscript are as follows:

1. Throughout the manuscript, 105K cells deficient in Tsc2 are used as a cancer model with hyperactive mTOR. It's clear this is a very stable model, but I am curious how the results hold in a more natural context of mTOR activation (i.e., growth factors or amino acid supplement) or under a more direct activation of mTORC1 like site mutagenesis for constitutive activation? Much of the paper is devoted to comparing Tsc2 KD to baseline, but overexpression or direct constitutive activation is more direct and recapitulate what happens in cancer.

Response: We agree that our Tsc2-deficient cells are stable models of hyperactive mTORC1. In new experiments, we investigated whether activation mutations of mTORC1 would change B7-H3 expression by expressing 4 different activated mTORC1 mutants in 293T cells. We found expression of all 4 mTORC1 mutants increased B7-H3 expression compared to wild-type mTORC1 (Fig. s1k). To further understand how B7-H3 is regulated in a more natural context of mTOR activation, we also performed new experiments where we stimulated wild-type mouse embryonic fibroblasts with amino acids and examined B7-H3 protein expressing by immunoblotting. We found amino acid stimulation leads to increased B7-H3 expression (Fig. s1l).

s1k

s1l

2. There is no mention of how JASPAR was used. What were the inputs, species, approach, dataset used, etc. As far as I can find, JASPAR only lists the CCAT motif for humans.

Response: We apologize that we neglected to include the use Genomatix MatInspector software in addition to JASPAR to predict for transcription factors that bind to CD276 promoter region. This is now included in the Result section (line 199-201).

3. In Figures 6d and 6f, MHC-II expression measured by lysate and IHC isn't entirely fair for evaluating antigen presentation. Extended data figure 5h essentially shows that the surface expression of MHC-II doesn't change. It also cannot be ignored that mTOR itself can induce antigen presentation.

Response: We thank the Reviewer for highlighting this. We do not and cannot claim the elevation of MHC-II expression in sh-B7-H3 tumors can be used to evaluate antigen presentation. Our conclusion is that inhibition of B7-H3 in the presence of an intact adaptive immune system can upregulate MHC-II expression. This is due to increased T-cell activity in the tumor milieu in mice carrying sh-B7-H3 tumors. We note that the experiments in Extended Data Fig 5h were done *in vitro* (in the absence of an immune system) which we believe is why a difference in MHC-II expression was not observed. We agree with the Reviewer that mTOR can induce antigen presentation in professional antigen presenting cells.

4. In all in-vivo tumor models, except for one (Extended Figure 3i), B7-H3 is either deleted before implantation or anti-B7-H3 treatment starts at the time of implantation. It would be useful to see more examples of B7-H3 inhibition in established tumors. The effectiveness of inhibition is expected to be most effective when the treatment is prophylactic, and the tumor hasn't been infiltrated by the immune system. On a related note, having an orthotopic model (either genetic or implantable) would show more breadth.

Response: We agree with the Reviewer about the importance of inhibiting B7-H3 in established tumors. In new experiments, we examined the efficacy of anti-B7-H3

antibody therapy in *Tsc2*^{+/-} mice, which develop kidney cysts and tumors. At 7 months of age, *Tsc2*^{+/-} mice were treated with isotype control or anti-B7-H3 antibody every 3 days for a total of 8 treatments (Extended Data Fig. 3m). Anti-B7-H3 antibody significantly reduced the burden of both gross and microscopic kidney tumors (Extended Data Fig. 3n, 3o). The gross kidney tumor score was decreased by 50% ($p < 0.001$) (Extended Data Fig. 3p) and the microscopic kidney tumor score was reduced by 70% ($p < 0.001$) (Extended Data Fig. 3q).

5. Without overinterpreting a UMAP, It is strange to see naive CD4 and CD8 T cells segregate so well, and yet the CD4 and CD8 cytotoxic T cell populations are directly adjacent to each other. Given that samples were processed at the same time and pooled, it may not have been necessary to integrate the data as it can overcorrect. Have you examined the data without integration to see if it is merited?

Response: As suggested, we reanalyzed the data without integration. We still see that naïve CD4⁺ and CD8⁺ T cells segregate well, and cytotoxic or effector CD4⁺ and CD8⁺ T cells are directly adjacent (see Fig below).

6. Figure 7p, the signature used on TCGA data is derived from scRNAseq of sorted T cells. Since tumor cells were not included in the process to generate the signature, the resultant genes could be present or aberrant in tumor cells. When calculating bulk tumor samples, the tumor cells themselves could affect the scoring independent of the actual cell quantities the authors wish to measure. It may be useful to take a ConsensusTME

approach and remove genes that have at $r > 0.25$ correlation with tumor purity before testing the score.

Response: As suggested, we extracted the TCGA tumor purity data from the ConsensusTME tool and analyzed the correlation of tumor purity with each of the 17 cytotoxic CD4⁺ T cells genes obtained from our sorted T cells CITE-seq data. We found that all the genes have a negative correlation with tumor purity (see Fig below); therefore we did not remove any genes for our analysis.

References

- 1 Hernandez, J. O. R. et al. A tissue-bioengineering strategy for modeling rare human kidney diseases in vivo. *Nat Commun* 12, 6496, doi:10.1038/s41467-021-26596-y (2021).
- 2 Henske, E. P. et al. Loss of heterozygosity in the tuberous sclerosis (TSC2) region of chromosome band 16p13 occurs in sporadic as well as TSC-associated renal angiomyolipomas. *Genes Chromosomes Cancer* 13, 295-298, doi:10.1002/gcc.2870130411 (1995).
- 3 Carbonara, C. et al. Apparent preferential loss of heterozygosity at TSC2 over TSC1 chromosomal region in tuberous sclerosis hamartomas. *Genes Chromosomes Cancer* 15, 18-25, doi:10.1002/(SICI)1098-2264(199601)15:1<18::AID-GCC3>3.0.CO;2-7 (1996).
- 4 Sepp, T., Yates, J. R. & Green, A. J. Loss of heterozygosity in tuberous sclerosis hamartomas. *J Med Genet* 33, 962-964, doi:10.1136/jmg.33.11.962 (1996).
- 5 Liu, H. J. et al. TSC2-deficient tumors have evidence of T cell exhaustion and respond to anti-PD-1/anti-CTLA-4 immunotherapy. *JCI Insight* 3, doi:10.1172/jci.insight.98674 (2018).
- 6 Tang, Y., Kwiatkowski, D. J. & Henske, E. P. Midkine expression by stem-like tumor cells drives persistence to mTOR inhibition and an immune-suppressive microenvironment. *Nat Commun* 13, 5018, doi:10.1038/s41467-022-32673-7 (2022).

REVIEWERS' COMMENTS

Reviewer #1 (Remarks to the Author):

The authors have addressed my previous critiques.

Reviewer #2 (Remarks to the Author):

The authors have satisfactorily addressed most of my previous questions with additional data. The only recommendation I have now is to add a brief discussion about how the discovery described in this report can potentially improve the current PD1/PDL1 blockade therapy, which has become the front-line therapy for a number of cancers.

Reviewer #3 (Remarks to the Author):

The authors replied to most of my questions. Although the nuclear translocation of STAT1 (Figure 6E) is still unclear, the manuscript has now significantly improved with respect to the previous version.

Reviewer #4 (Remarks to the Author):

In this manuscript, Liu and colleagues present compelling data suggesting mTORC1 activity is sufficient to maintain CD276 expression in tumor cells through indirect activation of the YY2 transcription factor. Subsequently, they show that a loss of CD276 abrogates tumor growth in-vivo due to a loss in immune evasion characteristics related to increased T cell infiltration, increased MHC-II expression, and activation of CD4 (and to some degree CD8) T cells within the microenvironment. This study includes very thorough mechanistic experimentation and, for the most part, appears relevant to a broad spectrum of cancers. In the prior review, the first figure had significant flaws and overall detracted from the importance of the question. The revisions made improved the figure and the impact of the paper as a whole. Below I address prior review comments and author rebuttals:

Reviewer #4 points 1-6 on figure 1 – The authors addressed all concerns presented and I have no issues with the results of the follow-up analyses. It is noteworthy that the hazards of CD276 and mTORC1 scores are largely not significant despite the individual scores having some significance in a 1-2 subtypes + overall, suggesting the two scores do not confer combinatorial hazards. Additionally, I appreciate the authors addressing the correlations in figure 1n and 1o in the text. I have no issue with the inclusion of the figures; however, I feel it does hurt the beginning of this story as 3 of these 4 correlations show no association (despite significant p-values, because they will be significant purely due to the number of datapoints).

Reviewer #4 other points 1-6 – The authors appropriately addressed all concerns. The additional studies are clear and support the claims made in the paper.

Concluding remarks:

Overall, this manuscript presents significant results for a number of biomedical fields and represents a leap in our mechanistic understanding of mTOR, B7-H3, and their coupled roles in immunity and cancer. The study presents a modest clinical story, with limited significance in TCGA data; however, considering the larger amount of molecular studies in this manuscript, the limited clinical application is sufficient to suggest a connection. I want to conclude by thanking the authors for an honest and thorough revision of their manuscript.

We are very appreciative of the Reviewers for their support and assistance in strengthening our manuscript for publication.

Our point-by-point responses are below.

Reviewer #1 (Remarks to the Author):

The authors have addressed my previous critiques.

Response: We are happy to have addressed the comments satisfactorily and thank the Reviewer for their constructive comments in the development of this manuscript.

Reviewer #2 (Remarks to the Author):

The authors have satisfactorily addressed most of my previous questions with additional data. The only recommendation I have now is to add a brief discussion about how the discovery described in this report can potentially improve the current PD1/PDL1 blockade therapy, which has become the front-line therapy for a number of cancers.

Response: As requested, we have included a brief discussion in the Discussion to describe how the data reported in this manuscript can potentially improve the current PD-1/PD-L1 blockade therapy (line 565-573). We thank the Reviewer for the time and contribution to the improvement and finalization of this manuscript. Please see below for the discussion we have included.

“The relationship between B7-H3 expression and PD-L1 expression may be clinically relevant. Although, multiple data sets indicate that PD-L1 is not upregulated at baseline in TSC-associated tumors, including our prior work using immunohistochemistry ¹ and our recent work using single cell RNA sequencing ². Interestingly, we examined PD-L1 (CD274) expression in our ex vivo sorted tumor cells and found that PD-L1 (CD274) is upregulated by 2.56-fold ($p_{adj} = 1.46E-10$) in sh-B7-H3 tumor cells compared to sh-NC (Extended Data Table 1). These data could provide a rationale for testing B7-H3 blockade in combination with PD-1/PD-L1 blockade therapy in tumors with mTORC1 activation.”

Reviewer #3 (Remarks to the Author):

The authors replied to most of my questions. Although the nuclear translocation of STAT1 (Figure 6E) is still unclear, the manuscript has now significantly improved with respect to the previous version.

Response: We thank the Reviewer for the positive comments. We have increased the dpi to 3000 for Figure 6E to try to further enhance the resolution.

Reviewer #4 (Remarks to the Author):

In this manuscript, Liu and colleagues present compelling data suggesting mTORC1 activity is sufficient to maintain CD276 expression in tumor cells through indirect activation of the YY2 transcription factor. Subsequently, they show that a loss of CD276 abrogates tumor growth in-vivo due to a loss in immune evasion characteristics related to increased T cell infiltration, increased MHC-II expression, and activation of CD4 (and to some degree CD8) T cells within the microenvironment. This study includes very thorough mechanistic experimentation and, for the most part, appears relevant to a broad spectrum of cancers. In the prior review, the first figure had significant flaws and overall detracted from the importance of the question. The revisions made improved the figure and the impact of the paper as a whole. Below I address prior review comments and author rebuttals:

Response: We thank the Reviewer for the encouraging comments about the compelling data, the thorough mechanistic experimentation, and the potential for relevance to a spectrum of cancers. As detailed below, we have worked hard to address further comments.

Reviewer #4 points 1-6 on figure 1 – The authors addressed all concerns presented and I have no issues with the results of the follow-up analyses. It is noteworthy that the hazards of CD276 and mTORC1 scores are largely not significant despite the individual scores having some significance in a 1-2 subtypes + overall, suggesting the two scores do not confer combinatorial hazards. Additionally, I appreciate the authors addressing the correlations in figure 1n and 1o in the text. I have no issue with the inclusion of the figures; however, I feel it does hurt the beginning of this story as 3 of these 4 correlations show no association (despite significant p-values, because they will be significant purely due to the number of datapoints).

Response: We agree about the lack of significance of the hazard scores of mTORC1 score and CD276 expression. We have removed figures 1n and 1o.

Reviewer #4 other points 1-6 – The authors appropriately addressed all concerns. The additional studies are clear and support the claims made in the paper.

Response: Thank you.

Concluding remarks:

Overall, this manuscript presents significant results for a number of biomedical fields and represents a leap in our mechanistic understanding of mTOR, B7-H3, and their coupled roles in immunity and cancer. The study presents a modest clinical story, with limited significance in TCGA data; however, considering the larger amount of molecular studies in this manuscript, the limited clinical application is sufficient to suggest a connection. I want to conclude by thanking the authors for an honest and thorough revision of their manuscript.

Response: We are extremely grateful for this positive evaluation of our work.

References

- 1 Liu, H. J. et al. TSC2-deficient tumors have evidence of T cell exhaustion and respond to anti-PD-1/anti-CTLA-4 immunotherapy. *JCI Insight* 3, doi:10.1172/jci.insight.98674 (2018).
- 2 Tang, Y., Kwiatkowski, D. J. & Henske, E. P. Midkine expression by stem-like tumor cells drives persistence to mTOR inhibition and an immune-suppressive microenvironment. *Nat Commun* 13, 5018, doi:10.1038/s41467-022-32673-7 (2022).